# LOGO — Long cOntext aliGnment via efficient preference Optimization

**Zecheng Tang** [1 2]  **Zechen Sun** [1 2]  **Juntao Li** [1 2]  **Qiaoming Zhu** [1 2]  **Min Zhang** [1 2]

## Abstract

Long-context models (LCMs) have shown great potential in processing long sequences, with research showing they can accurately locate token-level salient information. Yet, the generation performance of these LCMs is far from satisfactory and might result in misaligned responses, such as hallucinations. To enhance the generation capability, existing works have investigated the effects of data size and quality for both pre-training and instruction tuning stages. Though achieving meaningful improvement, previous methods fall short in either effectiveness or efficiency. In this paper, we introduce LOGO, an efficient and effective training strategy that first introduces preference optimization for long-context alignment. LOGO consists of a reference-free preference optimization strategy and a corresponding efficient data synthesis process. By training with only 0.3B data on a single 8×A800 GPU machine for 16 hours, LOGO allows the Llama-3-8B-Instruct-80K model to achieve comparable performance with GPT-4 in real-world long-context tasks while preserving the model's original capabilities on other tasks, e.g., language modeling and MMLU. Besides, LOGO can also scale the models' context window size while enhancing their performance.

## 1. Introduction

Processing long input sequences is a fundamental capability for Large Language Models (LLMs) nowadays, with a few models being able to process context lengths exceeding millions of tokens (Team et al., 2024; MiniMax et al., 2025). This capability unlocks the potential of LLMs for novel tasks and cutting-edge applications, such as high-resolution image processing (Tian et al., 2024) and long video understanding (Weng et al., 2025). Additionally, long-context models (LCMs) eliminate the need for complex toolchains and intricate workflows, e.g., RAG (Yu et al., 2024), that were previously required to address context-length constraints (Ravaut et al., 2024), facilitating tasks such as long-document summarization (Laban et al., 2024) and project code analysis (Zhu et al., 2024).

Yet, recent studies have pointed out that open-source LCMs failed to achieve satisfactory performance in real-world or complex synthetic long-context tasks, where LCMs might produce misaligned results, such as instruction unfollowing and hallucinations (Tang et al., 2024; Zhang et al., 2024a). To mitigate the above issues, the open-source community has made significant strides, primarily focusing on building and scaling up high-quality synthetic data to post-tune the models (Wu et al., 2024a; Bai et al., 2024; Fu et al., 2024; Li et al., 2024). These efforts, as shown in figure 1, have led to notable improvements but still fall short in terms of either effectiveness or efficiency. For instance, the Llama-3.1-8B-128K model (AI@Meta, 2024a), trained on over 800B long-instruction data, still underperforms the Llama-3-8B-80K model (Zhang et al., 2024b), which was post-trained with only 1.5B high-quality data. However, the Llama-3-8B-80K model shows only slight improvement over the backbone and still lags far behind closed-source models like GPT-4.

To investigate the reasons behind LCMs generating misaligned outputs, we visualized the information retrieval capability (reflected by Retrieval Score) and the generation capability (reflected by Recall Score) of different LCMs on a synthetic retrieval task[1]. As shown in figure 1(b), we can observe a minimal difference among the retrieval scores of various LCMs, but a large difference in their generation performance. This suggests that **while LCMs are adept at identifying key information within long contexts, they struggle to effectively utilize the retrieval information for generation**. The underlying cause might stem from the fact that these methods primarily focus on optimizing the training data for the Supervised Fine-Tuning (SFT) stage, which aims to enhance the adherence of LCMs to long instructions

---

[1]School of Computer Science and Technology, Soochow University [2]Key Laboratory of Data Intelligence and Advanced Computing, Soochow University. Correspondence to: Juntao Li <ljt@suda.edu.cn>.

*Proceedings of the 42$^{st}$ International Conference on Machine Learning*, Vancouver, Canada. PMLR 267, 2025. Copyright 2025 by the author(s).

---

[1]Retrieval capability is reflected through the recall score of salient tokens located by retrieval heads (Wu et al., 2024b). We calculate the average recall score across the top-10 retrieval heads. A higher retrieval score indicates that the LCM can retrieve more critical information. Details are shown in Appendix A.

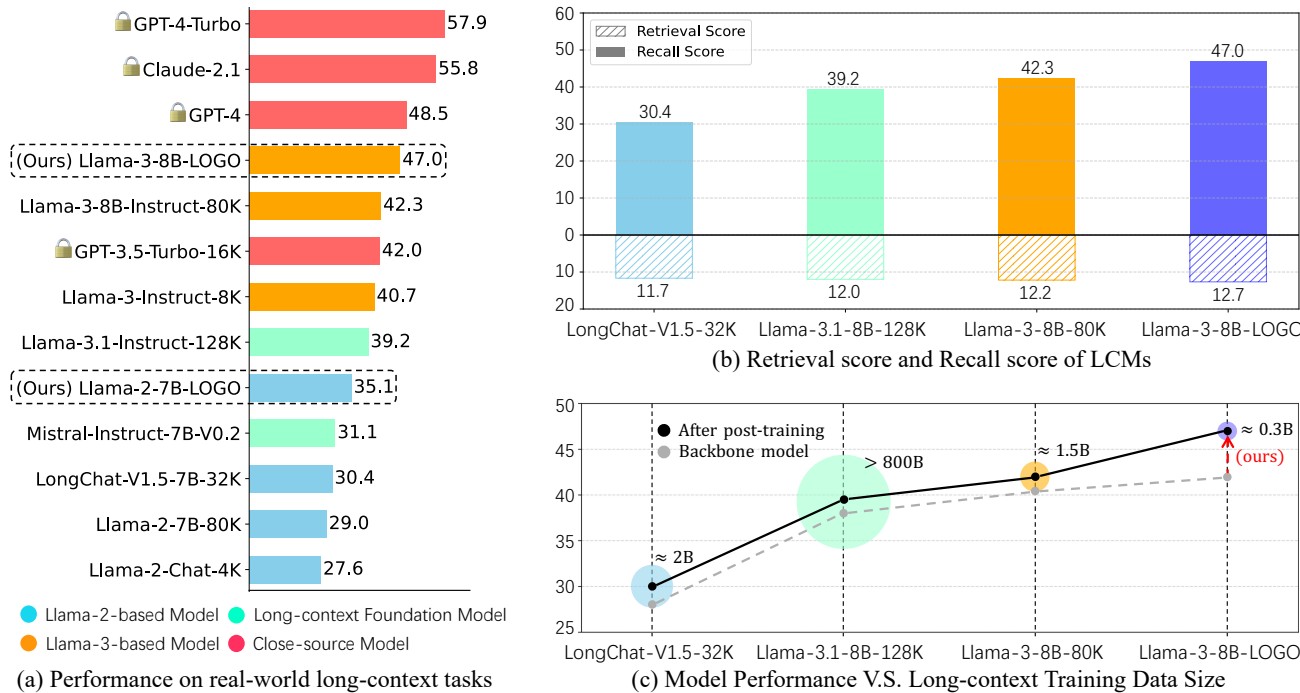

Figure 1. (a) Performance of widely-used LCMs on real-world long-context tasks, where some LCMs feature large context window size but relatively poor performance; (b) Retrieval score (information retrieval ability) and Recall score (generation ability) of LCMs on the synthetic retrieval long-context task (multi-value NIAH); (c) Long-context (pre-)training data size for each LCM.

but lack further preference-alignment training to further calibrate the model's outputs. Additionally, given that the sequence length of the context is typically much longer than the prediction portion, the feedback signal during the SFT stage, i.e., cross-entropy loss, from the prediction is often overshadowed by that from the context. Thus, merely optimizing LCMs with inadequate SFT data is insufficient.

In this paper, we propose **LOGO**, an efficient and effective **L**ong c**O**ntext ali**G**nment training strategy based on preference **O**ptimization that is suitable for both context window scaling and long-context alignment. Two key components are involved in LOGO: (1) an efficient preference data synthesis pipeline tailored for long-context scenarios, and (2) a long-context preference optimization training objective. It is worth noting that training with long sequences is a memory-intensive task (Dao, 2023) and the DPO algorithm also has a high GPU memory demand. Thereby, LOGO adopts a reference-free training objective and the positional indices synthesis method (Zhu et al., 2023) to overcome the GPU memory-bound and improve the training efficiency.

By training with LOGO, LCMs can achieve significant improvements in real-world tasks and gain moderate improvements in synthetic and language modeling tasks, as well as maintaining good performance on the short-context tasks, e.g., MMLU (Hendrycks et al., 2020). As shown in figure 1(a), our Llama-3-8B-LOGO significantly outperforms

GPT3.5-Turbo in real-world tasks and approaches the performance of some top closed-source models like GPT-4. Additionally, LOGO can also benefit the context window scaling stage of short-context LLMs such as Llama-2-7B-Chat-4K (Touvron et al., 2023), where we can extend their context window size up to 8 times (e.g.,32K context window size for Llama-2-7B-Chat-4K) and simultaneously enhancing their performance substantially.

## 2. Related Work

### 2.1. Context Window Scaling and Long-context Alignment

Two steps are essential for empowering LLMs with the ability to handle long-context tasks: 1) context scaling, which expands the limited context window size to support long-context tasks, e.g., from 8k to 128k; and 2) long-context alignment, which ensures that LCMs can follow long instructions. Currently, the open-source community mainly focuses on the former, primarily by (1) post-training models on long instruction data (Chen et al., 2023b; Xiong et al., 2023; Fu et al., 2024; Zhang et al., 2024b), (2) devising novel model architectures (Yang et al., 2023; Zhang, 2024; Tworkowski et al., 2024), and (3) modifying positional encoding (Peng et al., 2023a; Chen et al., 2023a; Jin et al., 2024) to extend the context window of LLMs. However, current works (Tang et al., 2024; Hsieh et al., 2024; Zhang

et al., 2024a; You et al., 2024) indicated that LCMs still underperform in long-context tasks, frequently manifesting issues such as hallucinations and failure to follow instructions, despite possessing large context window size. To mitigate this issue, Bai et al. (2024); Wu et al. (2024a); Chen et al. (2024) proposed to align the LCMs in long-context scenarios by synthesizing long-dependency instruction data to fine-tune the models. Some LLMs are even pre-trained with massive long instruction data (Jiang et al., 2023; Dubey et al., 2024; Abdin et al., 2024). Yet, despite numerous attempts that have been made to improve the data quality and quantity, the performance of open-source LCMs still lies far behind close-source LCMs. In this work, we tackle the above challenge by rethinking the long-context training objective. We introduce an efficient long-context preference optimization training strategy, i.e., LOGO. With a small amount of data and computational resources, LOGO can significantly enhance the model performance of LCMs.

## 2.2. Direct Preference Optimization

Direct Preference Optimization (DPO) (Rafailov et al., 2024) is a widely adopted RLHF algorithm (Ouyang et al., 2022) that aims to align models with human preferences. Compared to other reinforcement learning methods, e.g., PPO (Schulman et al., 2017), DPO can achieve strong performance while eliminating the need for a separate reward model. Different from Supervised Fine-Tuning (SFT) which models to align with ground truth at the token level, DPO updates model parameters based on discrete evaluation scores. Specifically, DPO guides the model to "reject" misaligned responses and "accept" preferred responses with differently assigned prediction scores. Significant efforts have been made to improve the DPO, such as RSO (Liu et al., 2023), CPO (Xu et al., 2024), TPO (Saeidi et al., 2024), and ORPO (Hong et al., 2024). Compared to short-context tasks, obtaining DPO data for long-context tasks is considerably more challenging due to the absence of open-source evaluation models designed for long-context tasks and the complexities involved in manual annotation. Thereby, we propose an efficient and theoretically guaranteed method for long-context DPO data synthesis.

## 3. Methodology

### 3.1. Background

**Direct Preference Optimization** Given prompt $x$, DPO aims to maximize the likelihood of a chosen response $y_w$ over a rejected one $y_l$, thereby preventing the model from generating undesired content. There are three essential modules in DPO: one frozen reference model and one trainable policy model for calculating the DPO loss jointly, and one evaluation strategy (or evaluation model) for distinguishing between $y_w$ and $y_l$. SimPO (Meng et al., 2024) is an im-

proved variant of DPO, which employs an implicit reward formulation that directly aligns with the generation metric, e.g., PPL, thereby eliminating the need for a reference model. The training objective of SimPO can be written as:

$$\mathcal{L}_{\text{Sim}}(\pi_\theta) = -\mathbb{E}_{(x,y_w,y_l)} \left[ \log \sigma \left( \frac{\beta}{|y_w|} \log \pi_\theta(y_w|x) \right. \right. \\ \left. \left. - \frac{\beta}{|y_l|} \log \pi_\theta(y_l|x) - \gamma \right) \right], \quad (1)$$

where $\pi_\theta$ is the policy model, $\beta$ (scaling of the reward difference) and $\gamma$ (target reward margin) are the hyper-parameters to separate the preferred and dis-preferred responses.

**Positional Indices Synthesis** Transformer-based models rely on positional indices to identify the relative position of each token (Raffel et al., 2020). One efficient method to extending the model's input context length is to adjust the positional indices, simulating long-sequence inputs without modifying the actual input sequence (Press et al., 2021; Ruoss et al., 2023). By default, the positional indices of a sequence of length $K$ are $\mathbb{P}(K) = \{0, 1, \cdots, K-1\}$. To extend the sequence length from $K$ to $K'$, we can synthesize the positional indices: $\mathbb{P}_{\mathcal{B}}(K') = \{0 + b_0, 1 + b_1, \cdots, K-1 + b_{K-1}\}$, where $\mathcal{B}$ indicates the positional bias $\{b_0, b_1, \cdots, b_{K-1}\}$ applied to each original position index and $K - 1 + b_{K-1} = K'$. To ensure effectiveness, the synthesis of position indices should achieve a uniform distribution of relative distances within the extended sequence length $[0, K']$ and cover as many of the extended positional indices as possible (Wu et al., 2024a).

### 3.2. Long-context alignment with LOGO

#### 3.2.1. TRAINING DATA SYNTHESIS OF LOGO

Given the challenges of preference data synthesis for long-context tasks, we begin by presenting the process of LOGO's preference data synthesis. For each long-context sample, we can format it as a triplet $\{Q, \mathcal{C}, P\}$, where $Q$, $\mathcal{C}$, and $P$ represent the question, the reference long-context, and the model prediction, respectively. As shown in Fig. 2, we first divide $\mathcal{C}$ into equal-length chunks $\{C_1, C_2, \cdots, C_n\}$. Then, three steps are involved: (1) Importance Scoring with Automatic Evaluator, (2) Preference Data Synthesis, and (3) Positional Indices Synthesis.

**Importance Scoring with Automatic Evaluator** To construct preference data in the long-context scenario, an efficient method is to guide the model to respond based on contexts with varying degrees of influence on the outcome. More concretely, to construct the chosen data, we only provide the model with context relevant to the question, thus enhancing the fidelity of the model's output by reducing contextual interference (Shi et al., 2023). Conversely, we

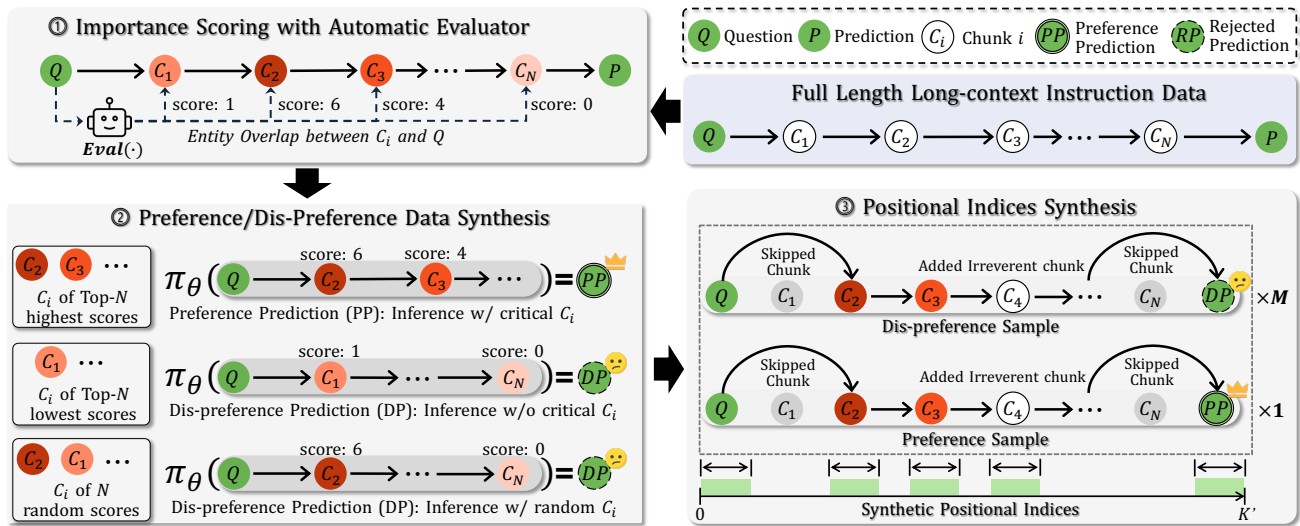

*Figure 2.* Training data synthesis process of LOGO, where we guide the model to generate preference data based on the detected critical segments within the long context and apply a positional synthesis strategy to "extend the sequence length".

utilize irrelevant context or reduce key information to guide the model in generating misaligned content like hallucinations. To find the relevant chunks $C_i$ within the context, we utilize an automatic evaluator $\text{Eval}(\cdot)$ to measure the "contribution" of each chunk $C_i$ to the question $Q$. Specifically, we identify all the entities within a chunk $C_i$ and calculate the overlapping entities between each $C_i$ and $Q$, where a greater overlap indicates a higher influence of $C_i$ to $Q$. Thereby, we can efficiently assign importance scores $sS = \{s_1, s_2, \cdots, s_n\}$ to all the chunks.

**Preference Data Synthesis**   Let $N$ represent the number of chunks within a context, we define a threshold $\delta$ to filter the critical chunks. Specifically, chunks $\mathbb{C}_{>\delta}$ scoring above $\delta$ are considered as critical chunks while chunks $\mathbb{C}_{<\delta}$ scoring below $\delta$ are considered as irreverent chunks. Then, we combine $Q$ and $\mathbb{C}_{>\delta}$ for model to generate correct prediction $P_{\text{chosen}}$, and adjust the ratio of chunks sampled from $\mathbb{C}_{>\delta}$ and $\mathbb{C}_{<\delta}$ for model to generate wrong predictions $P_{\text{rejected}}$.

It is worth noting that the open-source community currently lacks effective methods to effectively identify models' misaligned outputs for the long-context tasks, which poses a great challenge for selecting chosen and rejected samples in preference data synthesis[2]. Therefore, **instead of constructing one rejected sample with a specific error pattern, we can expand the rejected space to push the model away from a range of possible wrong predictions**. Specifically, we construct $P_{\text{rejected}}$ based on two misaligned error patterns: (1) model generation based on all irrelevant chunks ($\mathbb{C}_{<\delta}$), and (2) model generation based on partially

relevant chunks ($\mathbb{C}_{<\delta} \cup \mathbb{C}_{>\delta}$). Subsequently, the constructed $P_{\text{chosen}}$ and $P_{\text{rejected}}$ share the same context $\mathcal{C}'$, which is constructed from all the chunks in $\mathbb{C}_{>\delta}$ and partial chunks in $\mathbb{C}_{<\delta}$. Finally, one LOGO training sample can be written as $\left( \{Q, \mathcal{C}', P_{\text{chosen}}\}, \{Q, \mathcal{C}', P_{\text{rejected}}^{(i)}\}_{i=1}^{M} \right)$, where $M$ is the rejected sample number in equation 4.

**Positional Indices Synthesis**   Given that each preference data includes $(M + 1)$ instances, with one chosen sample and $M$ rejected samples, a long context length of $|\mathcal{C}'|$ can easily lead to GPU out of memory (even with 80GB memory). To address this, we employ a positional encoding synthesis strategy. By assigning different synthetic positional indices to each chunk, we can simulate long-sequence input with just short context (Wu et al., 2024a). Specifically, to ensure that the synthetic positional indices do not disrupt the semantic structure of short context, the positional indices within each chunk should be continuous, while indices between adjacent chunks can be discrete, i.e., omitting certain positional indices (as shown in sub-Fig. ③ in Fig. 2). Given $N$ equal-length chunks within each sample[3], to achieve a uniform distribution of relative distance within the expanded context length $[0, K']$, each positional bias term $b_i \in \mathcal{B}$ should be sampled from a uniform distribution. The synthetic positional indices can be written as:

$$\begin{cases} \mathcal{P}_{\mathcal{B}}(K) = \{i + b_i\}_{i=0}^{k-1}, \\ \text{where } b_i \sim \mathcal{U}(1, (i \bmod |C_i|) \times (K' - K)/N), \end{cases} \quad (2)$$

---

[2]We analyze the method of utilizing GPT-4 as an evaluation model and show some misalignment cases in Appendix B.

[3]Since the length of question $\mathcal{Q}$ and prediction $P$ are much shorter compared to the long context $\mathcal{C}$, we can ignore the length of $\mathcal{Q}$ and $\mathcal{P}$ for simplicity.

where $(i \bmod |C_i|)$ indicates the chunk index where the current positional index $i$ resides, and $(K' - K)/N$ represents the expansion size for each chunk.

We provide a theoretical guarantee that the synthetic positions can cover all possible scenarios given an adequate amount of data. Further details on the positional synthesis process are provided in Appendix E.

### 3.2.2. TRAINING OBJECTIVE OF LOGO

The training objective of LOGO can be written as:

$$\mathcal{L}_{\text{LOGO}}(\pi_\theta) = \mathcal{L}(\pi_\theta) + \lambda \mathbb{E}_{(x,y_w)} \log \pi_\theta(y_w|x)), \quad (3)$$

where $\mathcal{L}(\pi_\theta)$ is the preference optimization term, $\lambda$ is the hyper-parameter, and $\mathbb{E}_{(x,y_w)} \log \pi_\theta(y_w|x))$ is the SFT regularization term[4].

As mentioned above, we expanded the rejected sample space to address the issue of lacking long-context evaluation models. Consequently, we guide the model to simultaneously reject multiple potential rejected samples to construct the preference loss objective. Thus, $\mathcal{L}(\pi_\theta)$ can be written as:

$$
\mathcal{L}(\pi_\theta) = -\mathbb{E}_{(x,y_w,y_l^{(1\cdots M)})} \left[ \log \sigma \left( \frac{\beta}{|y_w|} \log \pi_\theta(y_w|x) \right. \right.
$$
$$
\left. \left. - \frac{\beta}{M|y_l|} \sum_{j=1}^{M} \log \pi_\theta(y_l^{(j)}|x) - \gamma \right) \right], \quad (4)
$$

where $M$ is the number of rejected samples, and the remaining terms are consistent with those defined in equation 1.

It is worth noting that $\mathcal{L}(\pi_\theta)$ is also free of the reference model and more aligned with the generation tasks[5], which is efficient for long-context training. We have theoretically proven that $\mathcal{L}(\pi_\theta)$ has a small generalization error and shows a reliable model performance on the unseen data in Appendix C and conduct the convergence property analysis from the gradient perspective in Appendix D.

## 4. Experiment

### 4.1. Settings

**Training Dataset Construction** We construct the synthetic datasets based on two corpora: (1) 4,000 instances sampled from long-llm-data (Zhang et al., 2024b), which includes reference contexts from multiple domains (e.g., biography, paper, *etc.*) and questions generated by GPT-4, covering tasks such as Single-Detail QA, Multi-Detail QA,

and Summarization; (2) 2,000 instances sampled from Red-Pajama (Computer, 2023) to mitigate forgetting, where we prompt the open-source LCM Qwen2-70B-Instruct (Yang et al., 2024) to generate questions for each instance. Then, we split each instance into equal-length chunks, with each chunk containing 512 tokens. To construct preference and dis-preference data, we use the spaCy model[6], a named entity recognition (NER) model that can identify all the entities within a context, as the evaluator $\text{Eval}(\cdot)$. We use the number of overlapping entities between each chunk $C_i$ and the question $Q$ as the importance score. We set the threshold $\delta$ as 6, chunk number $N$ as 16, and $|C_i| = 512$. For positional indices synthesis, we adopt two different sampling strategies on positional bias $\mathcal{B}$ to ensure that all positional indices are uniformly covered and maintain the semantic structure of the context. Finally, we have a total number of 12,000 training samples, with a total data size of approximately $12{,}000 \times 512 \times 16 \times 3 \approx$ **0.3B tokens.**

**Training Details** To validate the effectiveness of our method while controlling experimental costs, we use the efficient fine-tuning method LoRA (Hu et al., 2021), which fine-tunes only the attention and token embedding modules. We set $M$ as 2 in equation 4. Due to the positional indices synthesis, LOGO can potentially scale the context window size and ensure alignment in long-context tasks simultaneously. To demonstrate the effectiveness of LOGO, we experiment with **two types of models**: (1) **Short-context Models (SCMs)** including Llama-2-7B-Chat (Touvron et al., 2023) and Llama-3-8B-Instruct (AI@Meta, 2024b), which own context lengths of 4K and 8K, respectively; and (2) **Long-context Models (LCMs)**, including Llama3-8B-Instruct-80K (Zhang et al., 2024b), Llama-2-7B-Instruct-80K (Fu et al., 2024) and Mistral-Instruct-7B-V0.2 (Jiang et al., 2023), which inherently have long context windows. For SCMs, we scale the context windows to 8 times their original context length. We set $\lambda = 0.1$ in Eq. 3 and search the hyper-parameters of equation 4 based on (Meng et al., 2024) for different models, where $\beta = 10, \gamma = 3$ for Llama-3-8B-based model, $\beta = 2.5, \gamma = 0.25$ for Mistral-Instruct-7B-V0.2-based model, and $\beta = 3, \gamma = 0.6$ for Llama-2-7B-based model. More training details are in Appendix F.

**Evaluation and Task Selection** We assess the LOGO training strategy across real-world long-context tasks and the synthetic retrieval task. Additionally, to explore the impact of LOGO training for short-context tasks, we evaluate models on MMLU (Hendrycks et al., 2020) and Truth-fulQA (Lin et al., 2021) tasks. For a comprehensive comparison, we evaluate LOGO against existing widely-used methods from two perspectives: (1) **direct context window scaling**, including YaRN (Peng et al., 2023a), PoSE (Zhu

---

[4]The regularization term serves two key purposes: (1) to prevent reward hacking (Yuan et al., 2024; Hong et al., 2024), and (2) to ensure that the policy model $\pi_\theta$ maintains its original capabilities acquired through SFT/pre-training without drifting away.

[5]Since $\mathcal{L}(\pi_\theta)$ is similar to the SimPO loss function.

[6]https://spacy.io/usage/models

*Table 1.* Evaluation results on LongBench benchmark. We comprehensively compare LOGO with different strategies, including Training-Free (Free), SFT and DPO. The comparison is also conducted under two settings: (1) context window scaling on SCMs and (2) long-context alignment on LCMs. LOGO achieves the best performance under all the settings.

| Models | Type | S-Doc QA | M-Doc QA | Summ | Few-shot | Synthetic | Avg. |
|---|---|---|---|---|---|---|---|
| GPT-3.5-Turbo-16K | - | 39.8 | 38.7 | 26.5 | 67.1 | 37.8 | 42.0 |
| GPT-4 | - | 45.1 | 55.0 | 28.3 | 72.3 | 41.8 | 48.5 |
| LongChat-v1.5-7B-32k | - | 28.7 | 20.6 | 26.7 | 60.0 | 15.8 | 30.4 |
| LLama-3.1-8B-Instruct-128K | - | 23.9 | 15.8 | 28.9 | 69.8 | 57.5 | 39.2 |
| **Results on SCMs** *(scaling ×8 context window)* | | | | | | | |
| Llama-3-8B-Instruct-8K | - | 39.3 | 36.2 | 24.8 | 63.5 | 39.9 | 40.7 |
|   + YaRN-64K (Peng et al., 2023b) | Free | 38.0 | 36.6 | 27.4 | 61.7 | 40.9 | 40.9 |
|   + PoSE-64K (Zhu et al., 2023) | SFT | 34.9 | 31.4 | 18.7 | 59.3 | 44.2 | 37.7 |
|   + LOGO-64K | DPO | **39.8** | **36.7** | **28.8** | **65.4** | **49.0** | **43.9** |
| Llama-2-7B-Chat-4K | - | 24.9 | 22.6 | 24.7 | 60.0 | 5.9 | 27.6 |
|   + Data-Engineering-80K (Fu et al., 2024) | SFT | **26.9** | **23.8** | 21.3 | **65.0** | 7.9 | 29.0 |
|   + LOGO-32K | DPO | 26.7 | 23.3 | **26.3** | 63.1 | **11.1** | **30.1** |
| **Results on LCMs** *(preserving original context window)* | | | | | | | |
| Llama-3-8B-Instruct-80K | - | 43.0 | 39.8 | 22.2 | 64.3 | 46.3 | 42.3 |
|   + LongLoRA (Chen et al., 2023b) | SFT | 39.3 | 36.2 | 26.8 | 63.5 | 48.0 | 42.8 |
|   + SimPO (Meng et al., 2024) | DPO | 43.2 | 40.7 | 23.5 | 66.7 | 48.4 | 44.5 |
|   + LOGO-80K | DPO | **44.0** | **41.2** | **28.1** | **68.6** | **53.0** | **47.0** |
| Llama-2-7B-64K | - | 28.3 | 33.2 | 13.4 | 62.3 | 6.1 | 28.7 |
|   + LongAlign (Bai et al., 2024) | SFT | 29.9 | 32.7 | 26.5 | 63.8 | 16.5 | 33.9 |
|   + LOGO-64K | DPO | **33.6** | **28.0** | **29.4** | **65.1** | **24.5** | **36.1** |
| Mistral-Instruct-7B-V0.2-32K | - | 31.7 | 30.6 | 16.7 | 58.4 | 17.9 | 31.1 |
|   + FILM-32K (An et al., 2024) | SFT | 37.9 | 34.9 | 25.3 | 64.7 | 31.2 | 38.8 |
|   + LOGO-32K | DPO | **38.3** | **37.6** | **26.1** | **67.0** | **31.5** | **40.1** |

et al., 2023) and data-engineering (Fu et al., 2024); and (2) **long-context alignment**, including LongLoRA (Chen et al., 2023b), LongAlign (Bai et al., 2024), and FILM (An et al., 2024). Due to space limitation, we report **more experimental results**, including model performance on language modeling task, results of scaling to longer context window size, and more short-context tasks in Appendix G.

### 4.2. Performance on Long-context Tasks

**Results on Real-world Long-context Tasks** We evaluate the LOGO performance on LongBench (Bai et al., 2023), a comprehensive benchmark suite encompassing 16 distinct datasets spread across 6 task categories, including Single Document QA (S-Doc QA), Multi-Document QA (M-Doc QA), Summarization (Summ), Few-shot, Synthetic, and Code. It is worth noting that we exclude the Code category since the code testing data primarily involves contexts of just around 4,000 tokens and our training data does not cover this domain. As shown in table 1, where we can observe that: (1) **LOGO achieves the best performance among all the subtasks** . Specifically, for SCMs, LOGO outperforms both training-free and fine-tuning methods. Although these methods can potentially extend the context window of SCMs, they significantly impair performance on real-world long-

context tasks. For instance, PoSE causes the Llama3-8B-Instruct model to drop around 3 points on average compared to the baseline, with particularly notable declines in performance on the summarization tasks. For LCMs, LOGO can significantly improve model performance, with all LCMs showing varying degrees of improvement, e.g., Llama-3-8B-Instruct-80K model shows an average 5-point improvement compared to the baseline, whereas the instruct tuning method tends to restrict even a well-performing LLMs to a limited performance bottleneck; (2) **Compared to other methods, LOGO demonstrates significant improvement in information-intensive tasks**, which require the model to gather information from various parts of the context. Specifically, in summarization and synthetic tasks, LCMs trained with LOGO achieve an improvement of at least 5 points.

**Evaluation Results on Synthetic Retrieval Task** To investigate whether the LOGO training strategy affects the information retrieval capabilities of LCMs, we conduct a Needle-in-a-Haystack (NIAH) testing (gkamradt, 2023). As shown in figure 3, we can find that LOGO can scale the context window for SCMs (left group) and does not adversely affect the original context window size of LCMs (right group). We can also observe that the original LCMs (middle group) and those trained with LOGO (right group) share similar

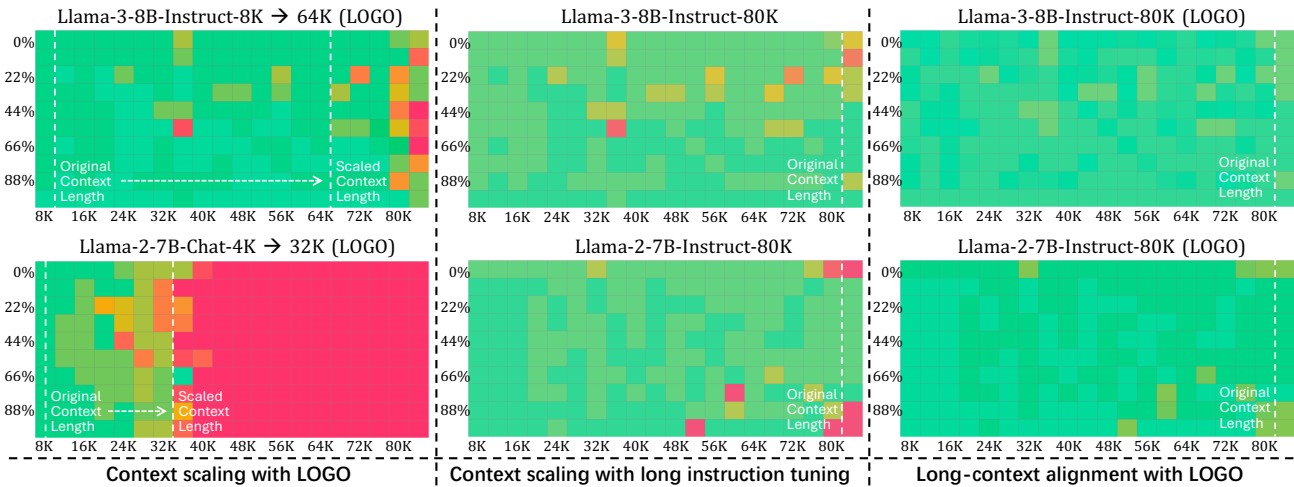

Figure 3. Results of the NIAH testing, where green and red indicate a successful and a failure recall, respectively. The test covers context lengths from 8K to 88K with incremental intervals of 0.5K. The needle is positioned at varying depths.

patterns, i.e., similar shades of color, yet LOGO improves performance in areas where the original LCMs fail. This indicates that LOGO does not compromise the inherent capabilities of LCMs but rather enhances their original weakness. We can also find that Llama-3-8B-8K model demonstrates superior context scaling results compared to Llama-2-7B-4K. This can be attributed to the larger RoPE base value of Llama-3-8B-8K (500,000) compared to that of Llama-2-7B-4K (10,000), which has been shown to facilitate context window scaling effectiveness (Su et al., 2023).

### 4.3. Performance on Short-context Tasks

MMLU (Hendrycks et al., 2020) and TruthfulQA (Lin et al., 2021) are two representative short-context tasks that assess LLMs' foundational capabilities. As shown in Fig. 4, we find that LOGO not only preserves the LLM's inherent capabilities on short-context tasks but also demonstrates improvements. This is because LOGO aims to teach the model to generate responses based on the context rather than fabricating results (such as producing hallucinations), which is equally applicable to short-context tasks. We can also find that scaling context length with LOGO yields better results than instruction tuning. For instance, in TruthfulQA task, Llama-3-8B-Instruct-80K shows significant performance degradation compared to our Llama-3-8B-LOGO. Such a phenomenon indicates a high "alignment tax" paid from instruction tuning (Fu et al., 2023).

## 5. Ablation Study

We analyze the impact of various hyper-parameters in the LOGO training objective in Sec. 5.1 and discuss the influence of positional indices synthesis in Sec. 5.2. We mainly utilize the Llama-3-8B-Instruct-80K model for the exper-

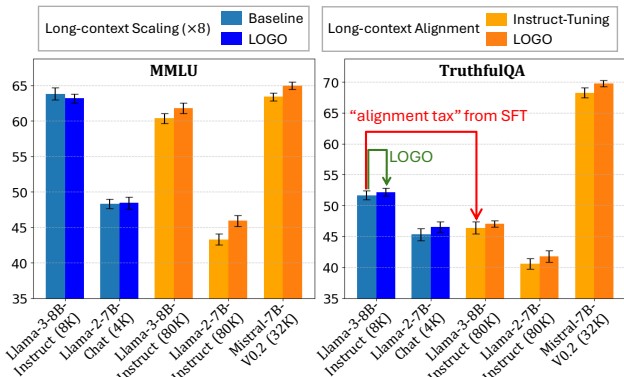

Figure 4. Model performance on short-context tasks, where the LOGO training objective can preserve the backbone model capability while long-context SFT might impair the model performance on short-context tasks.

iment, as it exhibits strong baseline performance across a wide range of tasks. The ablation study focuses on real-world tasks, reporting the average score on LongBench (LB), and evaluates language modeling performance by calculating the perplexity (PPL) score on the PG19 test set (Rae et al., 2019a) with a context length of 64K. Due to space limitation, we put the detailed analysis between the LOGO and the SFT training strategies in Appendix H and highlight the efficiency superiority of LOGO training in Appendix I.

### 5.1. Analysis of LOGO Training Objective

**Effect of SFT Regularization Term** $\lambda$   To investigate the impact of SFT regularization term in equation 3, we adjust the value of $\lambda$ from 0.0 to 1.0. As shown in figure 5(a), we present a scatter plot visualizing the model's performance under different $\lambda$ settings. We observe that increasing $\lambda$

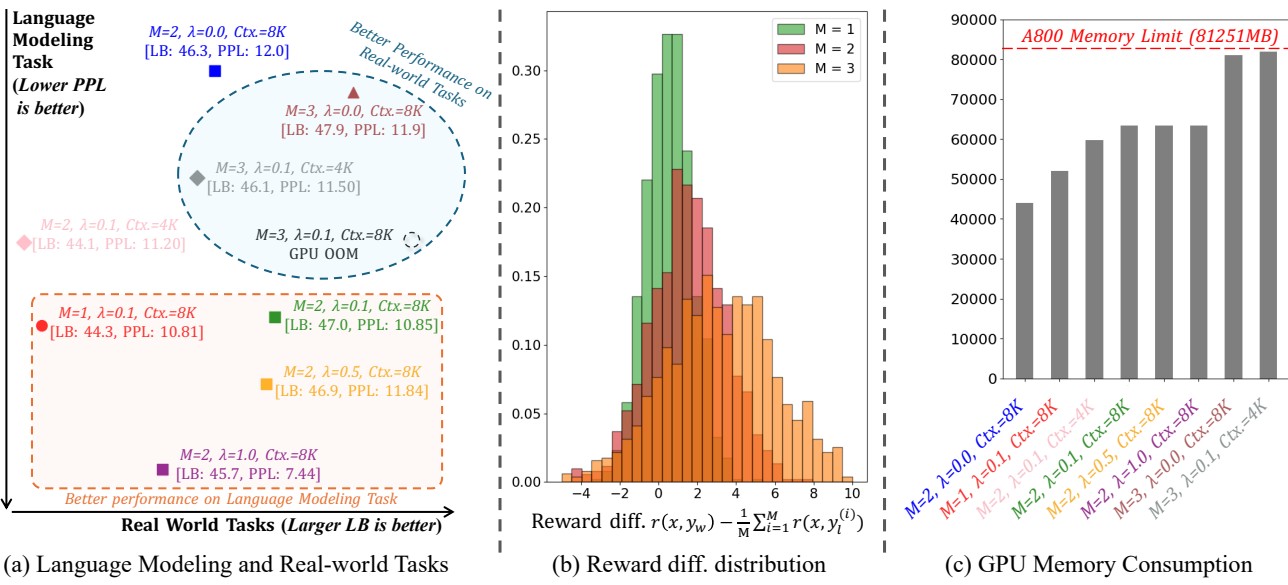

(a) Language Modeling and Real-world Tasks    (b) Reward diff. distribution    (c) GPU Memory Consumption

*Figure 5.* (a) Comparison among different settings on the language modeling task (PPL) and real-world tasks (Avg. score on LB); (b) Reward difference distribution under different $M$ settings; (c) Training GPU memory consumption of different settings.

allows the model to achieve a lower PPL score, while the change of $\lambda$ value have a minimal impact on performance for real-world tasks. For example, for the three sets of experimental results at $(M = 2, \lambda = 0.1, Ctx. = 8K)$, $(M = 2, \lambda = 0.5, Ctx. = 8K)$, and $(M = 2, \lambda = 1.0, Ctx. = 8K)$, we can observe that as $\lambda$ gradually increases, the PPL score decreases significantly by nearly 3.5 points, whereas the average score on LongBench differs by only around 1.5 points. This indicates that LOGO is robust to the SFT regularization term, and incorporating it can enhance the model's language modeling capability.

**Impact of the Rejected Samples Number** We experiment with different rejected sample numbers $M = \{1, 2, 3\}$ in equation 3. Specifically, when $M$ equals 1, the LOGO Objective degenerates into the SimPO objective. As shown in figure 5(a), for real-world tasks, more rejected samples can enhance the model capability; while for the language modeling task, it slightly affects the model performance. We also visualize the learned reward margin $r(x, y_w) - \frac{1}{M}\sum_{i=1}^{M} r(x, y_l^{(i)})$ under various $M$ values in figure 5(b). We can observe that using a larger $M$ can flatten the distribution and make it easier for the model to distinguish between preference and rejected samples as the gap between $r(x, y_w)$ and $\frac{1}{M}\sum_{i=1}^{M} r(x, y_l^{(i)})$ gradually increases with larger $M$. This is because increasing $M$ can cover more samples with various misalignment patterns. However, as illustrated in figure 5(c), increasing the value of $M$ presents a challenge, as it may exceed GPU memory constraints. It is worth noting that striking a balance between computational efficiency and performance gains

is crucial. We extend the LOGO with more rejected samples by applying the context parallelism strategy, i.e., ring-attention (Liu et al., 2024), and provide a detailed analysis of this computation-performance trade-off in Appendix I.

### 5.2. Analysis of Positional Indices Synthesis

Positional Indices Synthesis aims to simulate longer sequence lengths by synthesizing positional indices while keeping the original input tokens unchanged. To assess the impact of different synthetic lengths on training performance, we consider two settings for synthetic data length: extending from a real input length of 4K to a target length of 64K ($Ctx. = 4K$) and from a real input length of 8K to a target length of 64K ($Ctx. = 8K$). We keep the chunk size $|\mathcal{C}_i|$ constant and set the number of chunks as 8 and 16 for the two respective settings. As shown in figure 5(a), short-context synthetic data length significantly diminishes the model's performance on both the language modeling task and real-world tasks (data point $(M = 2, \lambda = 0.1, Ctx. = 4K)$ versus data point $(M = 2, \lambda = 0.1, Ctx. = 8K)$), but can still overcome the instruction tuning method (42.8 average score on LongBench) and effectively reduces the GPU memory requirement during training (figure 5(c)). This is because when the original context length is relatively small (4K), it requires scaling up by a larger factor (16 times) to reach the desired context length (64K). **With the same amount of data**, some positional indices may be missed or infrequently activated, thereby affecting performance.

# 6. Conclusion

In this paper, we find that commonly used training approaches for LCMs, i.e., SFT with a limited amount of data, may degrade the model's generation capabilities, leading to misaligned outputs. To mitigate this issue, we propose LOGO, a novel and efficient preference optimization strategy for both context window scaling and long-context alignment. We tackle the challenge of data scarcity in long-context preference optimization by introducing a Positional Indices Synthesis strategy for long-context preference data synthesis. Additionally, we design an efficient and effective preference optimization objective tailored for long-context alignment. With a single 8×A800 GPU machine and just 16 hours of training, LCMs can a achieve significant improvement in long-context tasks while preserving their inherent capabilities with the LOGO training strategy. Theoretical analysis and comprehensive experiments across various settings have validated the effectiveness of our method.

## Impact Statement

This paper introduces a novel **L**ong-c**O**ntext ali**G**nment strategy via preference **O**ptimization, **LOGO**. Currently, the long-context model community primarily relies on using larger amounts of pre-training data during the context window scaling phase and fine-tuning models with existing mixed long-text instruction data for long-context alignment. However, the benefits of the above approach remain limited. The field still lacks open-source and effective methods for long-context alignment, particularly in the areas of long-preference data synthesis and the development of efficient alignment training objectives. In this paper, during the preference data construction phase, there still remains a significant challenge due to the absence of suitable evaluation models to assess whether the outputs of LCMs are accurate or contain hallucinations. While utilizing higher-quality datasets, such as those created through manual annotation, could improve outcomes, we acknowledge this as an area for further improvement. As an academic paper, we have demonstrated the generalizability of our method through comprehensive experiments and theoretical proofs. We hope that this work will provide valuable insights to the open-source long-context model community, raise awareness of the importance of long-context alignment, and inspire future research with broader societal and technical impacts.

## Acknowledgements

We want to thank all the anonymous reviewers for their valuable comments. This work was supported by the National Science Foundation of China (NSFC No. 62206194), the Natural Science Foundation of Jiangsu Province, China (Grant No. BK20220488), the Young Elite Scientists Sponsorship Program by CAST (2023QNRC001), and the Priority Academic Program Development of Jiangsu Higher Education Institutions.

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

# A. Details of Preliminary Experiments

In this section, we illustrate the details of the preliminary studies in section 1, including the experimental settings, task definitions, and the retrieval score calculation.

**Experimental Settings** In figure 1(a) and figure 1(b), we evaluate the model performance on the subsets in LongBench (Bai et al., 2023), including Single Document QA, Multi-Document QA, Summarization, and Few-shot tasks. For each long-context model, we utilize the same official instructions to guide the model prediction.

**Multi-values Needle-in-a-Haystack** In figure 1(c), we calculate the retrieval score on the Multi-values Needle-in-a-Haystack dataset, which requires LCMs to recall multiple values within the context. We provide an example in figure 6:

---

**Multi-values Needle-in-a-Haystack**

*Context:*
... context ...
The best thing to do in San Francisco is to eat a sandwich and sit in Dolores Park.
... context ...
The best thing to do in New York is to eat a sandwich and visit the Statue of Liberty.
... context ...

---

*Question:*
What is the single best thing to do in both San Francisco and New York?

---

*Ground Truth:* (preference)
eat a sandwich

---

*Figure 6.* Demonstration of Multi-values Needle-in-a-Haystack testing sample.

The formal definition of the task is as follows: Given $n$ questions $vq$ and its corresponding answers $K = \{vk_j\}_{j=1}^n$ (the needle), we insert $K$ in a synthetic context $c$ (the haystack) at random position index ranges $P = \{vp_i\}_{i=1}^n$. We then require the models to answer $q$ based on the haystack with the inserted needle. It is worth noting that $q$ and $K$ are unique and irrelevant to the context, ensuring that if an answer is correctly generated, it is indeed copied from the context, not from the model's internal knowledge.

**Calculation of Retrieval Score** Based on Wu et al. (2024b), we define the retrieval score as the recall score of salient tokens located by retrieval heads. To enhance comprehension, we manage to utilize familiar symbols and definitions that align closely with previous research. Specifically, denote the current token being generated during the auto-regressive decoding process as $x$, and the attention score of a head as $a \in \mathcal{R}^{|c|}$. In the task of Multi-values Needle-in-a-Haystack, an attention head $h$ is denoted as a retrieval head if it meets the following criteria:

- $x \in \boldsymbol{k}_i$, where $\boldsymbol{k}_i \in K$ and $x$ is a token within any one of the needle sentences in $K$.

- $\boldsymbol{c}_j = x$, $j = \arg\max(\boldsymbol{a})$, $j \in \boldsymbol{p}_i$, $\boldsymbol{p}_i \in P$, i.e., the input token that receives the highest attention probability by this head is a token within any one of the needle in $K$ and is the same token as the currently generated token.

Let $\boldsymbol{g}_h$ be the set containing all copy tokens (also can be viewed as the located tokens) and pasted by a given head $h$, we define:

$$\text{Retrieval score for head } \ h = \frac{|\boldsymbol{g}_h \cap \boldsymbol{k}_i|}{|\boldsymbol{k}_i|}, \tag{5}$$

It is worth noting that the retrieval score represents a token-level recall rate of the most attended tokens by an attention head. After obtaining the retrieval score for each head, we start by filtering out the non-retrieval heads by setting the threshold at 0.1. This means that if a head performs copy-paste 10% of the time or more, it will be considered a retrieval head. Then, we calculate the retrieval head score by averaging the scores of the top 10 attention heads from the remaining retrieval heads.

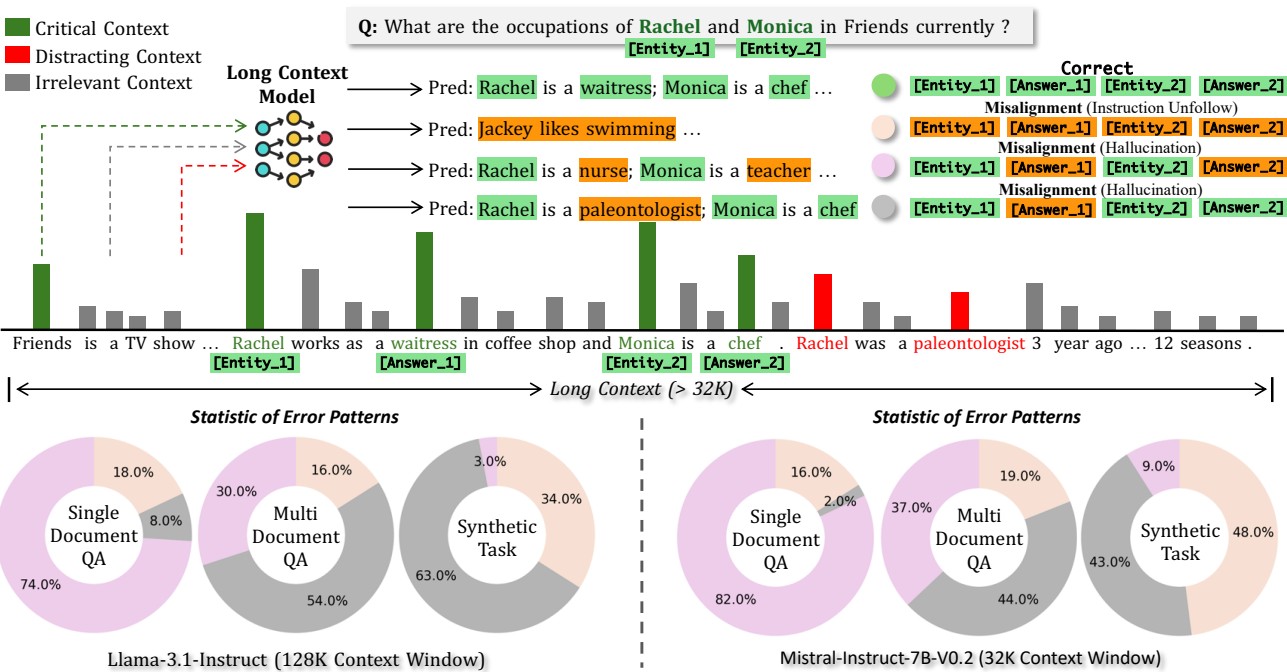

*Figure 7.* Demonstration and statistical analysis of different error patterns in long context tasks, where we have the following definitions of misalignment: (1) Instruction Unfollow: The entities in the model's prediction are different from the entities in the question; (2) Hallucination: The entities in the prediction overlaps with the entities in the question, but the answer is incorrect.

## B. Design of LOGO Training Objective and Error Pattern Definition in LCMs

Misaligned predictions generated from LCMs can be specifically categorized into two types: failing to follow instructions and generating hallucinations. In figure 7, we illustrate these two error patterns. Specifically, we define different error patterns by utilizing the degree of overlap between entities in the responses and the questions, along with specific templates:

- **Instruction Unfollow**: the entities in the model's responses do not overlap with the entities in the question.
- **Hallucination**: there is a partial intersection of entities between the model's responses and the question, and the entities in the response coincide with the main subject of the question.

It is worth mentioning that merely utilizing Named Entity Recognition (NER) models and rule-based methods proves inadequate for identifying these patterns. Instead, a more robust evaluation involving strong LLMs such as GPT-4 or human assessment is required to accurately identify these patterns. Consequently, in the design of the LOGO training objective, we do not confine to constructing cases with specific error patterns. Therefore, instead of finding one rejected sample with a specific error pattern, we can expand the rejected space to push the model away from a range of possible rejected samples.

## C. Error Bound Analysis

In this section, we analyze the error bound of the LOGO loss function (equation 4):

$$\mathcal{L}_{\text{LOGO}}(\pi_\theta) = -\mathbb{E}_{(x,y_w,y_l^{(1\cdots M)})} \left[ \log \sigma \left( \frac{\beta}{|y_w|} \log \pi_\theta(y_w|x) - \frac{\beta}{M|y_l|} \sum_{j=1}^{M} \log \pi_\theta(y_l^{(j)}|x) - \gamma \right) \right],$$

**Explanation of the Components in LOGO**

- **Sigmoid Function**: The sigmoid function $\sigma(z) = \frac{1}{1+e^{-z}}$ maps real-valued inputs to the range (0, 1), and the log-sigmoid function $\log \sigma(z)$ encourages large positive values of $z$.

- **Model Probabilities**: $\pi_\theta(y_w|x)$ is the probability of the preferred response given input $x$, and $\pi_\theta(y_l^{(j)}|x)$ is the probability of the dis-preferred response.

- **Scaling Factors** $\frac{\beta}{|y_w|}$ and $\frac{\beta}{M|y_l|}$ aim to normalize the log-probabilities by the lengths of the responses to account for varying lengths and scale the contribution using $\beta$. $\gamma$ is a margin hyper-parameter to ensure that the difference between preferred and dis-preferred responses exceeds a certain threshold.

The LOGO function aims to maximize the difference between the (normalized) log-probabilities of the preferred response and the average of the dis-preferred responses beyond a margin $\gamma$. Specifically, it encourages:

$$\frac{\beta}{|y_w|} \log \pi_\theta(y_w|x) - \frac{\beta}{M|y_l|} \sum_{j=1}^{M} \log \pi_\theta(y_l^{(j)}|x) \geq \gamma.$$

**To analyze the error bound, we can proceed by considering the following steps**:

**1. Bounding the Loss Function**

- **Upper Bound**: Since $\log \sigma(z) \leq 0$ for all real $z$, the negative log-sigmoid loss $-\log \sigma(z) \geq 0$.

- **Lower Bound**: The function $-\log \sigma(z)$ increases without bound as $z \to -\infty$, leading to potentially infinite loss values. However, in practice, model probabilities $\pi_\theta(y|x)$ are bounded below by a small positive value due to numerical stability (e.g., using softmax outputs and adding a small $\epsilon$).

**2. Assuming Bounded Log-Probabilities**   Let's assume that there exists a constant $C > 0$ such that:

$$-\log \pi_\theta(y|x) \leq C, \quad \forall y, x, \theta.$$

This assumption is reasonable since $\pi_\theta(y|x) \geq \epsilon > 0$ for numerical stability.

**3. Bounding $z$**   Given the boundedness of $-\log \pi_\theta(y|x)$:

$$\left| \frac{\beta}{|y_w|} \log \pi_\theta(y_w|x) \right| \leq \frac{\beta C}{|y_w|},$$

$$\left| \frac{\beta}{M|y_l|} \sum_{j=1}^{M} \log \pi_\theta(y_l^{(j)}|x) \right| \leq \frac{\beta C}{|y_l|}.$$

Thus, $z$ is bounded:

$$|z| \leq \beta \left( \frac{C}{|y_w|} + \frac{C}{|y_l|} \right) + |\gamma|.$$

**4. Lipschitz Continuity of the Loss Function**   The function $-\log \sigma(z)$ is Lipschitz continuous with Lipschitz constant $L = \frac{1}{4}$ since:

$$\left| \frac{d}{dz} \left( -\log \sigma(z) \right) \right| = \left| \frac{e^{-z}}{1 + e^{-z}} \right| = \frac{1}{1 + e^z} \leq \frac{1}{2}, \quad \forall z \in \mathbb{R}.$$

**5. Applying Concentration Inequalities**   Since the loss function is Lipschitz continuous and the losses are bounded, we can apply concentration inequality, i.e., McDiarmid's Inequality, to bound the difference between the empirical loss and the expected loss.

Let $\{(x_i, y_{w,i}, y_{l,i}^{(1 \cdots M)})\}_{i=1}^{N}$ be $N$ i.i.d. samples. Define the empirical loss:

$$\hat{\mathcal{L}}_{\text{LOGO}}(\pi_\theta) = \frac{1}{N} \sum_{i=1}^{N} \left[ -\log \sigma(z_i) \right],$$

where $z_i$ is the $z$ corresponding to the $i$-th sample.

McDiarmid's Inequality states that for all $\epsilon > 0$:

$$P\left(\mathcal{L}_{\mathrm{LOGO}}(\pi_\theta) - \hat{\mathcal{L}}_{\mathrm{LOGO}}(\pi_\theta) \geq \epsilon\right) \leq \exp\left(\frac{-2N\epsilon^2}{\sum_{i=1}^{N} c_i^2}\right),$$

where $c_i$ is the maximum change in the loss function due to the replacement of the $i$-th sample.

**6. Determining the Bounded Differences** $c_i$    Since the loss function change is bounded due to the boundedness of $z$ and the Lipschitz continuity:

$$c_i = \frac{1}{2} \cdot \left|z_{\mathrm{new},i} - z_{\mathrm{old},i}\right|,$$

where $z_{\mathrm{new},i}$ and $z_{\mathrm{old},i}$ are the values of $z$ before and after the change in the $i$-th sample.

Given the boundedness of $\log \pi_\theta(y|x)$ and the response lengths, we have a finite $c_i$.

**7. Bounding the Generalization Error**    Using the inequality, we can bound the probability that the empirical loss deviates from the expected loss by more than $\epsilon$:

$$P\left(\left|\mathcal{L}_{\mathrm{LOGO}}(\pi_\theta) - \hat{\mathcal{L}}_{\mathrm{LOGO}}(\pi_\theta)\right| \geq \epsilon\right) \leq 2\exp\left(\frac{-2N\epsilon^2}{\sum_{i=1}^{N} c_i^2}\right).$$

This inequality provides a *high-probability bound* on the generalization error—the difference between the expected loss and the empirical loss decreases as $N$ increases.

**8. Error Bound in Terms of Sample Size and Variability**    The error bound depends on:

- **Sample Size** $N$: Larger $N$ leads to tighter bounds.

- **Variability** $c_i$: Smaller $c_i$ (less variability in the loss function) leads to tighter bounds.

**Takeaway**    Based on the above analysis, we can get:

- To achieve a small generalization error, we need a sufficiently large sample size $N$. **In this paper, there are 0.3B tokens (6,000 samples) for training, which is enough for convergence**.

- Ensuring that the model probabilities $\pi_\theta(y|x)$ do not assign extremely low probabilities (avoiding numerical instabilities) to keep the loss function bounded and the $c_i$ small. **This is achieved by adopting the strong LLMs (e.g., Llama-3) for training**.

This analysis assures that with sufficient data and proper control of the model probabilities and response lengths, the loss function $\mathcal{L}_{\mathrm{LOGO}}(\pi_\theta)$ will have a small generalization error, leading to reliable model performance on unseen data.

## D. Convergence Property of LOGO from Gradient Analysis Perspective

In this section, we analyze the convergence property of the LOGO training objective from the gradient analysis perspective.

## D.1. Gradient Analysis

We first compare the gradient among three training objectives, i.e., DPO, SimPO, and LOGO. The gradient of those three training objectives can be written as:

$$\nabla_\theta \mathcal{L}_{\text{DPO}}(\pi_\theta) = -\beta \mathbb{E}_{(x, y_w, y_l) \sim \mathcal{D}} \left[ d_\theta \cdot \left( \underbrace{\nabla_\theta \log \pi_\theta(y_w | x)}_{\text{increase likelihood on } y_w} - \underbrace{\nabla_\theta \log \pi_\theta(y_l | x)}_{\text{decrease likelihood on } y_l} \right) \right],$$

$$\nabla_\theta \mathcal{L}_{\text{SimPO}}(\pi_\theta) = -\beta \mathbb{E}_{(x, y_w, y_l) \sim \mathcal{D}} \left[ s_\theta \cdot \left( \underbrace{\frac{1}{|y_w|} \nabla_\theta \log \pi_\theta(y_w | x)}_{\text{increase likelihood on } y_w} - \underbrace{\frac{1}{|y_l|} \nabla_\theta \log \pi_\theta(y_l | x)}_{\text{decrease likelihood on } y_l} \right) \right], \tag{6}$$

$$\mathcal{L}_{\text{LOGO}}(\pi_\theta) = -\mathbb{E}_{(x, y_w, y_l^{(1 \cdots M)}) \sim \mathcal{D}} \left[ l_\theta \cdot \left( \underbrace{\frac{1}{|y_w|} \log \pi_\theta(y_w | x)}_{\text{increase likelihood on } y_w} - \underbrace{\frac{1}{M|y_l|} \sum_{j=1}^{M} \log \pi_\theta(y_l^{(j)} | x)}_{\text{decrease likelihood on } y_l^{(1, \cdots, M)}} \right) \right].$$

where

$$d_\theta = \sigma \left( \beta \log \frac{\pi_\theta(y_l | x)}{\pi_{\text{ref}}(y_l | x)} - \beta \log \frac{\pi_\theta(y_w | x)}{\pi_{\text{ref}}(y_w | x)} \right),$$

$$s_\theta = \sigma \left( \frac{\beta}{|y_l|} \log \pi_\theta(y_l | x) - \frac{\beta}{|y_w|} \log \pi_\theta(y_w | x) + \gamma \right), \tag{7}$$

$$l_\theta = \sigma \left( \frac{\beta}{M|y_l|} \sum_{j=1}^{M} \log \pi_\theta(y_l^{(j)} | x) - \frac{\beta}{|y_w|} \log \pi_\theta(y_w | x) + \gamma \right).$$

In terms of gradient weight computation, SimPO and LOGO are similar in that they do not rely on a reference model. Instead, both methods calculate gradient weights based on the policy model itself. On the one hand, for both SimPO and LOGO training objectives, weight $s_\theta$ is higher for samples where the model incorrectly assigns a higher likelihood to the dis-preferred output $y_l$ or $y_l^{(1 \cdots M)}$, thereby focusing on correcting the model's mistakes. On the other hand, by considering multiple negative samples $y_l^{(1, \cdots, M)}$, LOGO enriches the learning signal of rejected samples. **This approach allows the model to learn a more comprehensive representation of undesirable outputs, improving its ability to reject a broader range of negative samples and helping the model to learn more patterns**.

## D.2. Convergence Properties

The combination of **self-contained gradient weights** and **length normalization** in LOGO promotes stable convergence. Since $l_\theta$ focuses on the policy model's own mispredictions without relying on a reference model, the learning process can adapt more freely based on the actual data, potentially leading to **faster and more robust convergence**. Besides, the use of a logistic loss function with a margin parameter $\gamma$ introduces non-linearity to the optimization problem. While the inclusion of multiple negative samples $y_l^{(j)}$ can provide a richer learning signal, and the **length normalization** helps in maintaining balanced updates, which can aid in convergence.

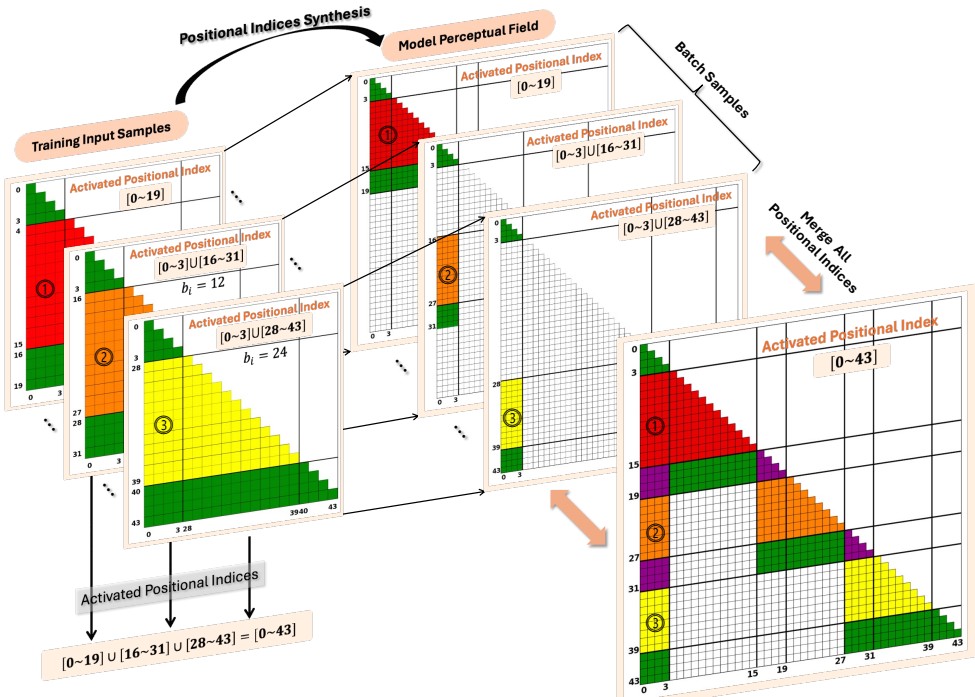

(a) Continuous Chunk Positional Indices Synthesis

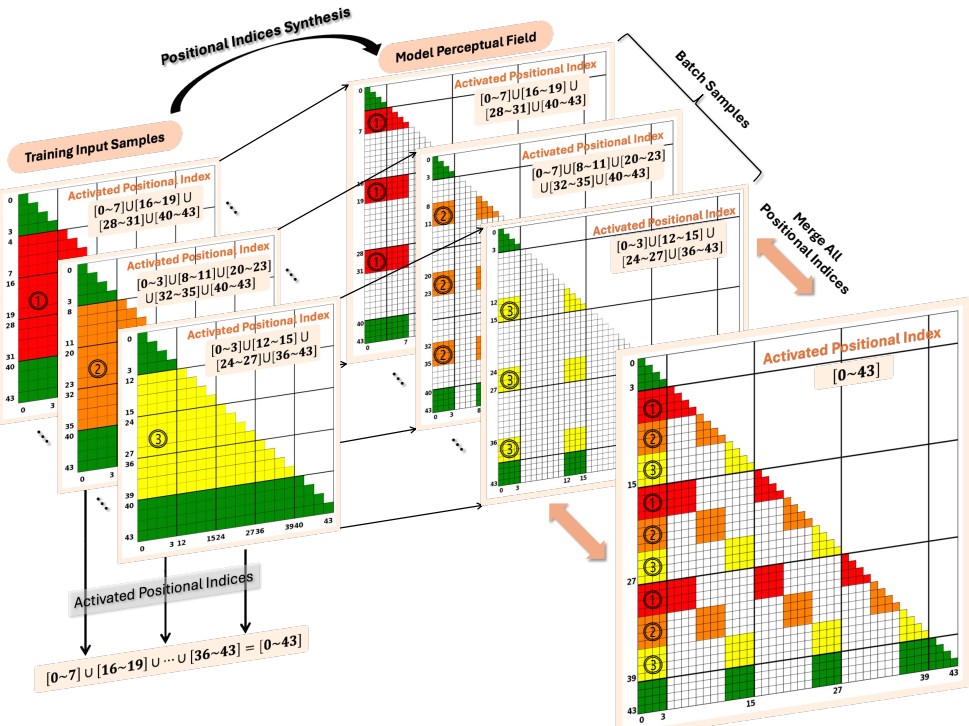

(b) Sparse Chunk Positional Indices Synthesis

*Figure 8.* Illustration of positional indices synthesis process, where the original context length is 19, and we extend it to a context length of 43. It is noteworthy that one batch in the figure corresponds to one training sample of LOGO, which contains one preference instance and several rejected samples.

# E. Positional Indices Synthesis

## E.1. Theoretical Guarantee of Positional Indices Synthesis

The theoretical guarantees for the position synthesis method can be equated to understanding why Relative Position Encodings (RPE) are effective. Existing works (Shaw et al., 2018; Zhu et al., 2023; Wu et al., 2024a) have already demonstrated that using positional indices synthesis can significantly improve training efficiency. Simply put, for the Transformer model, the model learns $(j - i)$, i.e., the difference between two positional indices, to understand the relative position information between any two tokens. **The context window scaling can be interpreted as the model's tendency to learn information about larger $(j - i)$ values**. Referring to the mathematical formulation of RoPE, the attention mechanism can be expressed as follows:

$$\text{Attn}_{i,j} = \frac{\left(QK^T + f(I, j)\right) V}{\sqrt{d}}$$

where $Q$, $K$, and $V$ are the query, key, and value hidden states, respectively.

To perform the length extrapolation, one only needs to consider how to make $(j - i)$ larger. The conventional approach is to increase the sequence length to enlarge the value of $(j - i)$, with each token corresponding to an absolute $i$ and $j$ value. As for position synthesis, it only needs to consider changing the values of $i$ and $j$ without altering the actual sequence length.

A potential issue is that some position indices may be missing in the positional synthesis method, and we illustrate how we compensate for the missing relative positions in the below section.

## E.2. Implementation Details of Positional Indices Synthesis

We visualize the positional indices synthesis process in figure 8. Specifically, to ensure that the synthesized positional indices do not disrupt the original text's semantic structure while maximizing the extended context size, we employ two different strategies for positional bias $\mathcal{B}$: Continuous Chunk Positional Indices Synthesis (figure 8(a)) and Sparse Chunk Positional Indices Synthesis (figure 8(b)). For Continuous Chunk Positional Indices Synthesis, the positional bias within the same chunk is consistent. For instance, in the first chunk $C_0$, the positional bias $\{b_0, b_1, \cdots, b_{|C_i|}\}$ are the same value sampled from distribution $\mathcal{U}(1, (K - k)/N)$.

This ensures that the semantic structure within the chunk remains intact but can lead to sparse synthesized positional indices, as there will be significant gaps between the positional indices among different chunks. Thereby, we propose Sparse Chunk Positional Indices Synthesis to fill these gaps, where each positional bias $b_i$ is sampled uniformly according to Equ. 2. Considering that Sparse Chunk Positional Indices Synthesis might disrupt the semantic structure of the text, we set the ratio of data for Continuous Chunk Positional Indices Synthesis and Sparse Chunk Positional Indices Synthesis to 9:1 in actual deployment.

# F. LOGO Training and Evaluation Details

To accelerate the training process and save GPU memory, we adopt DeepSpeed Zero 3 (Aminabadi et al., 2022). All the experiments are conducted on a $8 \times$ A800 (80GB) GPU machine, and all the training experiments are completed within 16 hours. We train all the models for two epochs, amounting to a total throughput of 0.3B * 2 = 0.6B tokens. The best model checkpoint is then selected based on performance on the validation set. For the training of the baseline, we followed the corresponding baseline settings.

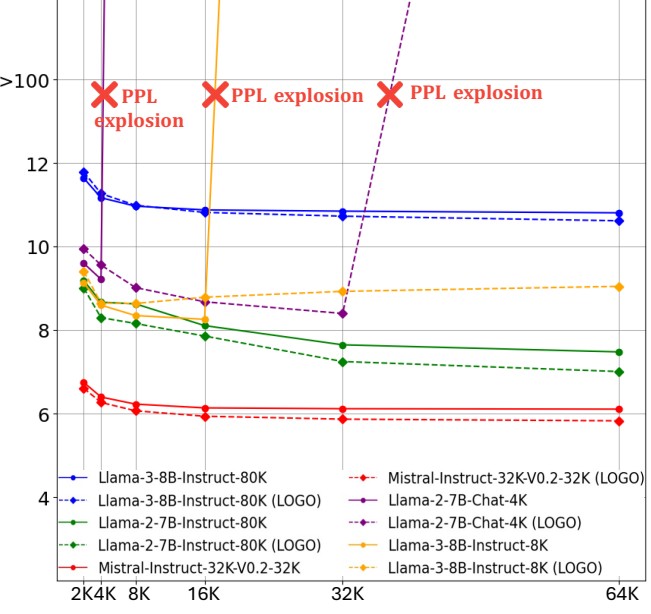

*Figure 9.* Evaluation results of language modeling task. The solid and dashed curves represent the PPL of the baselines and LOGO.

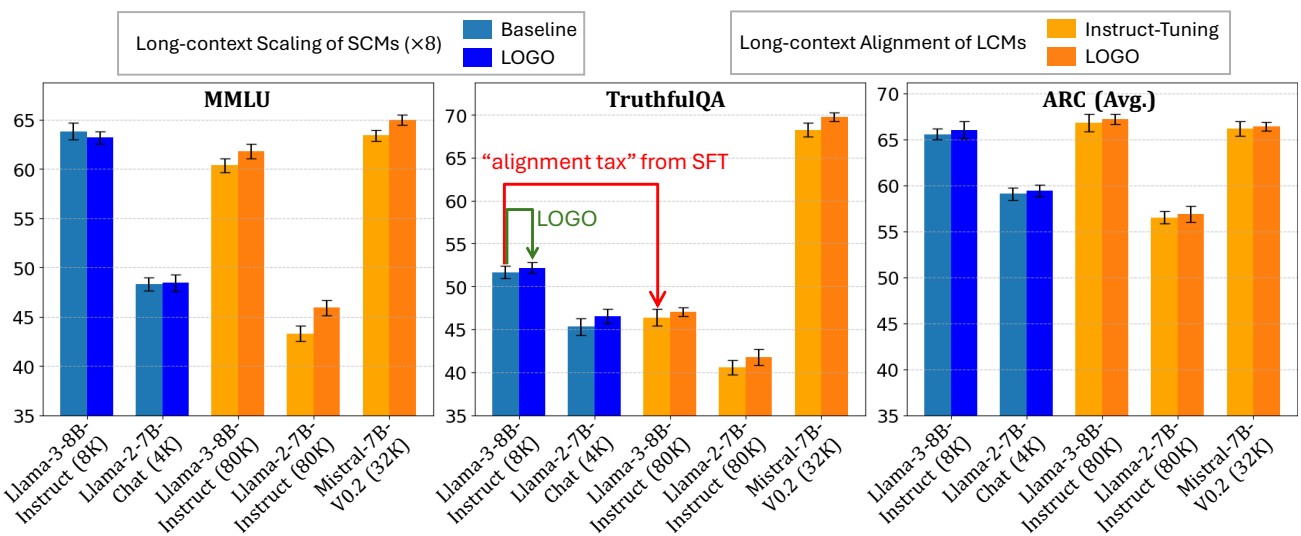

*Figure 10.* Model performance on short-context tasks, including MMLU, TruthfulQA, and ARC.

When testing on LongBench, we used the officially recommended context length of 32K (Bai et al., 2023), truncating any input sequence that exceeds this length.

## G. More Experimental Results of LOGO

### G.1. Evaluation Results on Language Modeling Task

We test the language modeling capability of LCMs by calculating the Perplexity (PPL) on the Gutenberg (PG-19) testing set (Rae et al., 2019b), with context lengths ranging from 2K to 64K. Considering that extremely long context lengths can cause the PPL calculation to exceed GPU memory, we apply the sliding window approach proposed by Press et al. (2021). As depicted in figure 9, for LCMs, such as Llama-3-8B-Instruct-80K and Llama-2-7B-Instruct-80K, using LOGO does not compromise the language modeling capability since the solid line (PPL of the backbone model) and the dashed line (PPL of LOGO) almost completely overlap. In the case of SCMs, such as the Llama-3-8B-Instruct-8K model, LOGO not only effectively scales the context window size of baseline models (the purple dotted curve versus the purple solid curve) but also achieves a lower PPL score compared to the SFT method since the yellow dotted curve (PPL of Llama-3-8B-Instruct-LOGO) is much lower than the blue solid curve (PPL of Llama-3-8B-Instruct-80K).

### G.2. Evaluation Results on Short-context tasks

Apart from the model results on MMLU (Hendrycks et al., 2020) and TruthfulQA (Lin et al., 2021), we also experiment with ARC (Hard and Easy) (Clark et al., 2018). The model performance is shown in figure 10, which is consistent with the conclusions presented in the main text.

### G.3. Scaling to Longer Context Window Size

We scale the context window size of the Llama-3-8B-Instruct-80K model from 80K to 256K and report the model performance in Table 2. Specifically, for longer context lengths, using only 0.3B tokens is insufficient to cover all position indices. Therefore, we expand the dataset size as the context length increases. We can observe that as we scale the context window size with more data, the model's performance also improves.

## H. Comparison between SFT and LOGO

As shown in figure 11, we illustrate the impact of SFT (with two loss calculation strategies following (Xiong et al., 2023)) and LOGO on the model's generation and understanding performance throughout the training process. We plot the trends of retrieval score (understanding ability) and recall score (generation ability) along the training progress. We can observe that

*Table 2.* Model performance when scaling to longer context window size.

| Model | Tokens | S-Doc QA | M-Doc QA | Summ | Few-shot | Synthetic | Avg. |
|---|---|---|---|---|---|---|---|
| Yi-6B-200K | - | 39.1 | 25.1 | 33.8 | 25.6 | 56.6 | 36.0 |
| Llama-3-8B-Instruct-LOGO-80K | 0.3B | 44.0 | 41.2 | 28.1 | 68.6 | 53.0 | 47.0 |
| Llama-3-8B-Instruct-LOGO-128K | 1.2B | 43.8 | 40.9 | 28.0 | 68.6 | 52.6 | 46.8 |
| Llama-3-8B-Instruct-LOGO-256K | 4.8B | **44.9** | **42.6** | **29.8** | **69.4** | **53.9** | **48.1** |

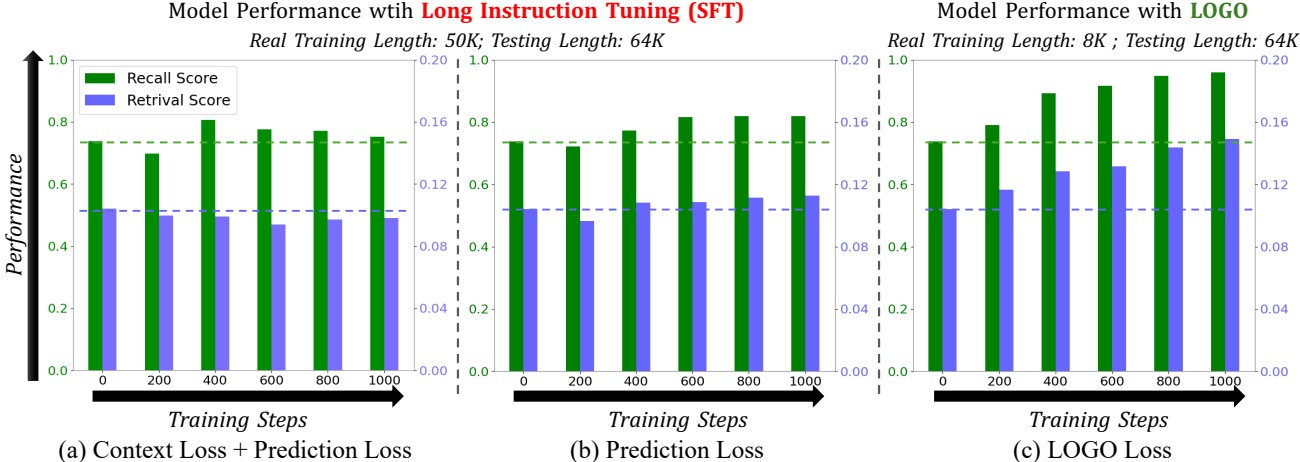

*Figure 11.* Comparison between SFT and LOGO training strategies on the synthetic retrieval task.

applying SFT loss to the entire sequence leads to a gradual decline in the LCM's understanding ability, accompanied by performance fluctuations; while applying SFT loss solely to the prediction portion shows no significant improvement in model performance. Nevertheless, applying LOGO can steer LCMs away from misaligned samples, thereby enhancing the recall score. Simultaneously, it improves comprehension abilities, enabling the model to retrieve more key information within the context.

## I. Trade-off between Training Computation and Model Performance

In this section, we present the trade-offs between computational efficiency and performance gains across different long-context alignment strategies. Specifically, we employed ring attention (Liu et al., 2024), a context-parallel approach, to implement training strategies for longer input sequences. We compared the following four strategies: (1) direct SFT training with the maximum input length supported by 80GB GPU memory (64K); (2) SFT combined with ring attention to train on longer sequences (128K); (3) our LOGO method; and (4) the LOGO method extended to longer training sequences in combination with ring attention. As shown in Table 3, we observe that extending the context length during training leads to slight performance improvements but comes at the cost of longer training times and reduced data throughput. For example, compared to LOGO, LOGO with Ring Attention achieves a 0.9-point improvement in performance. However, the throughput on 8 GPUs is only half of the original training setup (since we set the context-parallel size as 2), resulting in a total training time of 30 hours. Thereby, we believe that utilizing a position synthesis strategy to process more diverse data within 30 hours would likely yield better results while maintaining a more affordable resource consumption.

## J. Case Study of LOGO Data

In this section, we provide the training samples built based on the LOGO training data construction pipeline as illustrated in Sec. 3.2.1. We present the training samples in figure 12, figure 13, figure 14, and figure 15, where the training data exhibits different error patterns (misalignments) in their rejected samples.

*Table 3.* Training resource statistics among different settings. For ring attention (denote with *), we set context parallel size as 2, i.e., every 2 GPUs share 1 training sample.

| Training Strategy | Memory/GPU | Bsz/GPU | Total Throughput (8 GPUs) | Actual Training Length | Training Time (2,000 steps) | Real-world Task | Language Modeling |
|---|---|---|---|---|---|---|---|
| SFT | 80GB | 1 | 8 samples | 64K | 14h | 43.2 | **6.6** |
| SFT + Ring Attention | 45GB | 1 | 4 samples* | 128K | 24h | 44.3 | **6.6** |
| LOGO + Ring Attention | 69GB | 3 | 12 samples* | 64K | 30h | **47.7** | 9.8 |
| LOGO | 64GB | 3 | 24 samples | 12K | 16h | 47.0 | 10.4 |

---

**Case Study 1**

***Context:***
... context ...
If class time is used to allow students to apply what they have learned in cooperation with their peers, then the Flipped Classroom Model encourages collaboration, inquiry, and problem solving.
... context ...

---

***Question:***
Why is it important for teachers to use student-centered activities in the flipped classroom model?

---

***Ground Truth:*** (preference)
To encourage communication, inquiry, research, problem solving, and collaboration.

---

***Wrong Prediction1:*** (Instruction Unfollowing)
How can teachers create student-centered learning experiences in the classroom? What are some possible challenges in creating student-centered learning experiences in the classroom?

---

***Wrong Prediction2:*** (Hallucination)
Blended learning in action: A practical guide toward sustainable change. Corwin. Please note that you need to have at least a basic subscription to access the full text of the references...

*Figure 12.* LOGO training samples: case1.

# K. Error Analysis of Model Response

We conduct an error analysis on the results generated from different models, as illustrated in Figures 16∼ 20. Specifically, we focus on comparing the generated results from three models: Llama-2-LOGO, PoSE (Zhu et al., 2023), and LongAlign (Bai et al., 2024), all of which commenced training based on the Llama-2-Chat model (Touvron et al., 2023). As demonstrated in the cases, we mark irrelevant content with wavy lines and relevant content with underlines. Our analysis reveals that LOGO can generate accurate responses without being influenced by distracting information. In contrast, the other two methods (PoSE and LongAlign) are susceptible to interference from irrelevant information, which leads to wrong outputs.

---

**Case Study 2**

*Context:*
... context ...
As always in quantum physics, the joint measurement on Alice's two photons, concerning their relative state, produces a truly random result, just one among several possibilities.
... context ...
Alice's joint measurement thus produces a random result. Depending on this result, Bob's photon will always produce the result that the initial photon would have produced if it had been measured in the same direction, or else it will produce the opposite result to the one that would have been produced by the initial photon.
... context ...

---

*Question:*
What is the significance of the energy correlation between the two infrared photons?

---

*Ground Truth:* (preference)
If one photon has an energy above the average, the other photon will necessarily have an energy below the average.

---

*Wrong Prediction1:* (Instruction Unfollowing)
Bell's game is a theoretical game where two players, Alice and Bob, make independent choices of the polarization state of two photons before measuring the correlations between their polarizations...

---

*Wrong Prediction2:* (Instruction Unfollowing)
None, since there is no mention of energy correlation between two infrared photons in the provided context. The context only discusses quantum nonlocality, Bell's game, and quantum teleportation....

*Figure 13.* LOGO training samples: case2.

---

**Case Study 3**

*Context:*
... context ...
What is the slope of the line containing points (2, –3) and (4, 3)? slope = 3
... context ...
A slope that goes from low to high is positive. A slope that goes from high to low is negative. A slope that goes straight across is 0.
... context ...

---

*Question:*
What is the slope of the line containing points (2, –3) and (4, 3)?

---

*Ground Truth:* (preference)
3

---

*Wrong Prediction1:* (Instruction Unfollowing)
Do you want to solve it step by step?

---

*Wrong Prediction2:* (Hallucination)
The question can be interpreted as the difference in y-values divided by the difference in x-values. That is, let y1 = –3 and y2 = 3. Then x1 = 2 and x2 = 4. The slope is: = 6

*Figure 14.* LOGO training samples: case3.

---

**Case Study 4**

*Context:*
... context ...
And then there was no more speaking of the old lady, the house with the candle and the woman who was like a candle until, in the last train, they were nearly home.
... context ...

---

*Question:*
Why does the narrator stop asking his father about the grapes and the grape-vine?

---

*Ground Truth:* (preference)
Because of the unpleasant encounter with the old lady at the house.

---

*Wrong Prediction1:* **(Hallucination)**
The narrator stops asking his father about the grapes and the grape-vine because his father is not responding to the questions. The narrator is trying to learn about the grape-vine and its whereabouts but his father is not willing to provide any information...

---

*Wrong Prediction2:* **(Hallucination)**
Blended learning in action: Because his fatherś eyes seemed to fill with water and he began to speak in a small voice. At the same time, the narrator himself felt like his heart had been "torn in half."...

*Figure 15.* LOGO training samples: case4.

---

**Error Analysis 1**

*Context:*
... context ...
AEM outpoerforms both LEM and DPEMM by 6.5 and 1.7 respectively in F-measure on the FSD dataset, and 4.4 and 3.7 in F-measure on the Twitter dataset. We can also observe that apart from K-means, all the approaches perform worse on the Twitter dataset compared to FSD, possibly due to the limited size of the Twitter dataset.
... context ...

---

*Question:* What baseline approaches does this approach out-perform?

---

*Ground Truth:* K-means, LEM, DPEMM.

---

*LOGO (Ours):* AEM outperforms both LEM and DPEMM.

---

*LongAlign:* The proposed approach outperforms the baseline approaches on all three datasets.

---

*PoSE-YaRN-96k:* LEM and DPEMM.

*Figure 16.* Error analysis of responses from different models.

---

**Error Analysis 2**

*Context:*
... context ...
Larkin I. Smith Larkin Irvin Smith (June 26, 1944 – August 13, 1989) was an American Congressman from Mississippi serving for seven months until he was killed in a plane crash in Perry County, Mississippi in 1989. Smith was born in Poplarville, Mississippi to Nona Orene Bounds and her husband Hezekiah K. Smith.
... context ...

---

*Question:* In which county was Larkin I. Smith born?

---

*Ground Truth:* Pearl River County, Mississippi.

---

*LOGO (Ours):* Pearl River County, Mississippi.

---

*LongAlign:* Poplarville, Mississippi.

---

*PoSE-YaRN-96k:* Pearl River County.

*Figure 17.* Error analysis of responses from different models.

---

**Error Analysis 3**

*Context:*
... context ...
Locked On (novel) Locked On is a techno-thriller novel written by Tom Clancy and Mark Greaney released on December 13, 2011.
... context ... In 1832, Fougeres lived in the rue de Navarin.
... context ...

---

*Question:* What type of media does Locked On and Tom Clancy have in common?

---

*Ground Truth:* Novel.

---

*LOGO (Ours):* Novel.

---

*LongAlign:* Book.

---

*PoSE-YaRN-96k:* Video games. Books. Film.

---

*Figure 18.* Error analysis of responses from different models.

---

**Error Analysis 4**

*Context:*
... context ...
Retired merchants; just now in love with art; have a country-house at Ville d'Avray, and ten or twelve thousand francs a year.At the third sitting pere Vervelle mentioned a fine gallery of pictures which he had in his country-house at Ville d'Avray–Rubens.
... context ...

---

*Question:* Where do Mr. and Mrs. Vervelle live?

---

*Ground Truth:* In a mansion in Ville-d'Avray.

---

*LOGO (Ours):* Ville d'Avray.

---

*LongAlign:* In the country.

---

*PoSE-YaRN-96k:* They live in the rue de Navarin.

---

*Figure 19.* Error analysis of responses from different models.

---

**Error Analysis 5**

*Context:*
... context ...
Thaddeus P. Mott...At the time of his death, he was also the last surviving son of the eminent surgeon Valentine Mott...Upon his death in 1865, Mott was interred at Green-Wood Cemetery in Brooklyn, New York.
... context ...
Beaulieu-sur-Loire (French pronunciation:literally Beaulieu on Loire) is a commune in the Loiret department in north-central France.
Two days later, Anthony Roberts was on the scene with a detachment of Philadelphia police.
... context ...

---

*Question:* Where was the place of death of Thaddeus P. Mott's father?

---

*Ground Truth:* New York.

---

*LOGO (Ours):* New York.

---

*LongAlign:* Beaulieu-sur-Loire.

---

*PoSE-YaRN-96k:* Anthony Roberts.

---

*Figure 20.* Error analysis of responses from different models.

