# OpenReview forum: "LOGO --- Long cOntext aliGnment via efficient preference Optimization"
_ICML.cc/2025/Conference — ICML 2025 poster_

### Official Review · Reviewer_6VnN · 2025-03-14

**Overall Recommendation:** 3

**Summary:**

The paper introduces LOGO, a novel and efficient preference optimization strategy designed for long-context alignment in large language models (LLMs). LOGO addresses issues of misaligned responses in long-context models (LCMs) by introducing:
- A Reference-Free Preference Optimization Strategy.
- Efficient Data Synthesis for Long-Context Preference Optimization.
- Positional Indices Synthesis.

**Claims And Evidence:**

NA

**Essential References Not Discussed:**

NA

**Experimental Designs Or Analyses:**

The authors have conducted extensive experiments to validate the effectiveness of LOGO. The results demonstrate that LOGO significantly improves long-context alignment while maintaining efficiency.

**Methods And Evaluation Criteria:**

Yes, the methods and evaluation criteria proposed in this paper appear well-justified and appropriate for the long-context alignment

**Other Comments Or Suggestions:**

NA

**Other Strengths And Weaknesses:**

Strengths:
- The writing is well-structured, with clear logic and readability, making the paper easy to follow.
- The proposed LOGO method is a novel and efficient approach to long-context alignment, demonstrating both effectiveness and scalability.

**Questions For Authors:**

- Applicability to Reasoning Tasks:
LOGO focuses on long-context alignment and preference optimization. However, recent advances, such as OpenAI's O1 and DeepSeek's R1, have shown that test-time scaling can significantly enhance reasoning ability. Have you evaluated whether LOGO can also improve the model’s reasoning capabilities, particularly in long-context reasoning tasks?
- Efficiency Gains vs. Traditional RLHF Approaches
How does LOGO compare against standard RLHF approaches like PPO in terms of GPU usage and training time? Would LOGO still be beneficial in a setting with ample computational resources?

**Relation To Broader Scientific Literature:**

This paper makes significant contributions to the broader scientific literature on long-context alignment in LLMs. Prior works have primarily focused on scaling context window sizes (e.g., through post-training on long-instruction data, novel architectures, or positional encoding modifications). However, research has shown that large-context windows alone do not guarantee alignment, as models still exhibit hallucinations and fail to follow instructions effectively.

**Theoretical Claims:**

This paper does not contain significant theoretical proofs to verify.

---

> ### Author Rebuttal · Authors · 2025-03-29
>
> Dear Reviewer 6VnN, thanks for your insightful comments and suggestions. Below is our detailed response.
>
> ---
>
> **[Question 1]** Reasoning Capability Evaluation
>
> **[Re]** Thanks for raising this important point. However, our primary objective is **not to enhance reasoning per se**, but rather to **mitigate misalignment issues in long-context responses**.
>
> Nevertheless, we have extended our experiments with LongBench V2 in this rebuttal, which is specifically designed for real-world long-context reasoning (`Reviewer icM3’s Concern 2`). Our results indicate **significant performance improvements even in reasoning tasks**, particularly in settings ''without chain-of-thought (CoT) prompting''. Notably, Llama3-8B-80K achieves an 8-point average improvement, and Qwen2.5-7B-Instruct  yields a 2.6 average point improvement, suggesting that LOGO training may **implicitly enhance reasoning capabilities**—potentially due to the presence of multi-hop data in our synthesis preference data.
>
> To achieve reasoning performance on par with OpenAI’s O1 or DeepSeek’s R1, stronger RL supervision signals, such as *Long-CoT*, would likely be required. Interestingly, we have recently been investigating this direction: by enabling LCM to actively search for and integrate key information from long-context inputs before generating responses, we observe substantial performance gains. However, this exploration falls beyond the scope of LOGO itself and is an avenue for future work.
>
> ---
>
> **[Question 2]** Efficiency Gains vs. Traditional RLHF Approaches
>
> **[Re]**
> A key challenge in applying traditional RLHF methods (e.g., PPO, DPO) to long-context modeling (LCM) is their heavy reliance on reward models, critical models, etc. For instance, PPO requires a reward model, a reference model, and a value model, while DPO depends on a reference model.
>
> However, in the long-context domain, there are no publicly available reward models or value models to facilitate such training.  To address this, LOGO introduces an alternative: importance scoring and a novel training objective to replace the need for an offline reward model. This significantly improves training efficiency and eliminates the dependency on additional models.
>
> Besides, among open-source works, we have found only one study similar to ours: LongPO, which follows a traditional DPO-based approach and requires an additional reference model for training. To provide a clearer comparison, we present the following table highlighting the key differences between SFT, LOGO and LongPO:
>
> | Training Strategy          | Memory/GPU | Bsz/GPU | Total Throughput (8 GPUs) | Actual Training Length | Training Time (2,000 steps) | Real-world Task | Language Modeling |
> |----------------------------|------------|---------|---------------------------|------------------------|-----------------------------|-----------------|-------------------|
> | SFT                        | 80GB       | 1       | 8 samples                 | 64K                    | 14h                         | 43.2            | 6.6               |
> | SFT + Ring Attention       | 45GB       | 1       | 4 samples*                | 128K                   | 24h                         | 44.3            | 6.6               |
> | LOGO (w/o ref model) + Ring Attention      | 69GB       | 3       | 12 samples*               | 64K                    | 30h                         | 47.7            | 9.8               |
> | LOGO (w/o ref model) | 64GB       | 3       | 24 samples                | 12K                    | 16h                         | 47.0            | 10.4              |
> | LongPO (**w/ ref model**)  | OOM        | 2       | 16 samples                | 64K                    | -                         | -            | -              |
> | LongPO (**w/ ref model**) + Ring Attention (ring_size=2) | 62GB  | 2       | 8 samples  | 64K                    | >24h             | 44.7            | 17.6              |
>
> We observed that without employing the **Ring Attention** strategy or a CP-parallel approach, LongPO cannot be deployed on one 80GB GPU due to the combination of long input sequences and the requirement for an additional reference model. Even with the Ring Attention strategy—where every 2 GPUs process a segment of the sequence in parallel, the training time for 2,000 steps exceeds 24 hours. Additionally, LongPO's performance is inferior to LOGO, as it only applies a simplistic preference data processing method, where shorter responses are treated as preferred replies.
>
> In short, even in settings with **ample computational resources**, LOGO may still remain beneficial due to its **scalability and reduced model dependency**, making it a more practical and efficient alternative to traditional RLHF methods for long-context alignment.
>
> ---
>
> We hope our responses have adequately addressed your concerns. If you have any further questions, please don’t hesitate to ask.

---

### Official Review · Reviewer_2obo · 2025-03-14

**Overall Recommendation:** 3

**Summary:**

The paper addresses the challenge that open-source long-context models (LCMs) struggle with generation quality in long-context tasks, despite having strong information retrieval capabilities. These models often produce misaligned results, such as hallucinations and instruction-following errors, leading to low recall scores.
Key questions addressed:
Existing approaches primarily focus on scaling context length by adding more training data at the supervised fine-tuning (SFT) stage, which mainly improves retrieval capabilities.
There is a significant gap between retrieval and generation capabilities in LCMs—while they can locate important information, they struggle to use it effectively.
Constructing long-context preference pairs is difficult and underexplored in the literature.
Proposed approach:
 The authors propose LOGO, which consists of three key components:
A long corpus is broken into chunks, and each chunk is evaluated based on the number of overlapping entities with the given question.
Preferred and dispreferred samples are generated by combining different chunk compositions and prompting the model to produce responses.
Position indices synthesis is used during training to ensure efficiency and fit within hardware constraints.
Main results:
LOGO uses only 0.3B tokens.
Achieves comparable performance to GPT-4 and LLaMA on LongBench.
Maintains strong performance on standard benchmarks like MMLU.
Includes ablation studies on hyperparameters such as the number of negative samples and the impact of SFT regularization on final results.

**Claims And Evidence:**

Significant improvement on LongBench: This claim is mostly valid. However, some comparisons could be more rigorous. For example, in Table 1, YaRN is compared with LOGO, but the version of YaRN used is training-free. A fairer comparison would include a trained version of YaRN, as prior work has shown that training with context extrapolation methods (like YaRN) improves performance.
Performance improvement on synthetic tasks (e.g., "needles" evaluations): Well-supported by experiments in Figure 3 and Section 4.2.
No degradation on short-context evaluations and reduced hallucinations: Clearly supported by results in Section 4.3.

**Essential References Not Discussed:**

N/A

**Experimental Designs Or Analyses:**

More details on the training process of competitor models would help clarify the fairness of comparisons. For instance:
LLaMA-2-7B-Chat-4K is compared against LOGO trained with Data-Engineering-80K, but the datasets have different token counts (5B tokens for Data-Engineering-80K vs. LOGO).
Key hyperparameters like learning rate, batch size, and their impact on LOGO's number of negative samples are not discussed.
SFT Regularization Robustness Claim (Section 5.1):
The paper states that LOGO is robust to the SFT regularization term, as perplexity drops while task performance remains stable.
However, perplexity may not be a reliable indicator of long-context performance, as noted in previous research.
Additionally, it is unclear where the reported perplexity values come from (e.g., from the validation dataset during training or an evaluation set).
The inverse trend between LongBench scores and perplexity suggests that more evaluations are needed to fully validate the model's improvement.

**Methods And Evaluation Criteria:**

The authors use LongBench to evaluate the model. While LongBench is a reasonable choice, other benchmarks like Ruler would also be valuable.
LongBench and LongBench-v2 truncate input sequences from the middle, which may lead to different behaviors compared to evaluations with full-context retention. For example, GPT-4o excels on LongBench-v2 but lacks retrieval strength on Ruler. Including evaluations that retain full-context length would make the claims more robust.

**Other Comments Or Suggestions:**

N/A

**Other Strengths And Weaknesses:**

The strengths:
Novel idea on constructing long-context preference data synthetically. Prior literature on this is relatively scarce.
Conducted adequate experiments with various prior methods including Data Engineering and Extrapolation methods.
Results on long and short context tasks and analysis on the effectiveness of the proposed method.
Details on implementations on modeling and framework level make reproduction easier for audience.
The weaknesses:
Some comparisons may not be fair enough. Ex. comparing LOGO with training free YARN.
Evaluations on more benchmarks like Ruler will be helpful in providing a full picture of the method.

**Questions For Authors:**

Can we see a breakdown of performance across different context lengths in LongBench?It would be insightful to observe how performance varies with context length. This would help clarify whether gains come from data quality rather than the proposed method itself. A visualization of scores over context length could provide better insights into its impact.

**Relation To Broader Scientific Literature:**

The paper has adequate coverage of prior methods for long-context training at the SFT stage.
However, literature on preference training for LCMs is scarce, making direct comparisons with LOGO difficult.
The authors could reference additional related work on topics like memory compression and attention sparsity.

**Theoretical Claims:**

Briefly went through the bound analysis in Appendix C and it seems to be valid but did not look into detailsBriefly reviewed the bound analysis in Appendix C—it appears valid, but I did not examine it in depth.

---

> ### Author Rebuttal · Authors · 2025-03-29
>
> Dear Reviewer 2obo, we sincerely appreciate your thorough review of our work and the detailed feedback provided!
>
> ---
>
> **[Concern 1]** A fairer comparison of YaRN
>
> **[Re]** We have conducted the experiment and found that incorporating YaRN indeed leads to further performance improvements:
>
> | Model                  | S-Doc QA | M-Doc QA | Summ | Few-shot | Synthetic | Avg.  |
> |------------------------|---------|---------|------|----------|-----------|------|
> | Llama-3-8B-Ins-8K     | 38.0    | 36.6    | 27.4 | 61.7     | 40.9      | 40.9 |
> | + YaRN-64K           | 39.8    | 36.7    | 28.8 | 65.4     | 49.0      | 43.9 |
> | + LOGO-64K           | 39.8    | 36.7    | 28.8 | 65.4     | 49.0      | 43.9 |
> | + LOGO + YaRN-64K    | **40.7** | **37.4** | **28.9** | **67.3** | **50.4** | **44.9** |
>
> ---
>
> **[Concern 2]** Adding evaluations with full-context retention
>
> **[Re]** We have conducted the experiments and provided the evaluation results on Ruler below.
>
> | Model                                | Length | niah_1-3 | niah_multikey_1-3 | niah_multivalue | niah_multiquery | vt    | cwe   | fwe   | qa_1-2 | Total |
> |--------------------------------------|--------|----------|-------------------|----------------|----------------|-------|-------|-------|--------|-------|
> | Llama-3-8B-Instruct-80K-QLoRA-Merged | 32K    | 100.0    | 99.7              | 87.2          | 87.2          | 94.6  | 27.7 | 91.9 | 62.3   | 81.3  |
> | + LOGO                                 | 32K    | 100.0    | 100.0             | 93.2          | 92.5          | 95.2  | 28.1 | 93.2 | 66.2   | 83.5 **(+ 2.2)**  |
> | Llama-3-8B-Instruct-80K-QLoRA-Merged | 64K    | 100.0    | 99.0              | 84.5           | 84.5           | 92.6 | 0.2  | 78.7 | 59.5   | 74.9  |
> | + LOGO                                 | 64K    | 100.0    | 100.0             | 90.6          | 88.2          | 93.5 | 1.5  | 86.2 | 62.8   | 77.8 **(+ 2.9)** |
> | Llama-3-8B-Instruct-80K-QLoRA-Merged | 128K   | 99.9     | 77.9              | 76.0          | 76.0          | 88.6 | 0.3  | 81.9 | 52.0   | 69.1  |
> | + LOGO                                 | 128K   | 100.0    | 93.8              | 81.2           | 79.2          | 89.6  | 1.3  | 82.2 | 58.4   | 73.2 **(+ 4.1)** |
>
> ---
>
> **[Concern 3]** Lack of training process and evaluation details
>
> **[Re]** Below, we provide more details:
>  - **Fairness of Comparisons in Training Competitor Models** We acknowledge the token count difference between Data-Engineering-80K (5B tokens) and LOGO. For evaluation, we used the open-source pretrained checkpoint of the Data-Engineering-80K model, but the token format varies from LOGO, complicating direct comparison. Nonetheless, LOGO outperforms Data-Engineering-80K despite using a smaller dataset (0.3B tokens), showing its efficiency.
> - **Hyperparameters and Negative Samples** We used a fixed learning rate of 5e-5, with a cosine decay scheduler (100 steps warmup from 1e-8). The training setup included 8× A800 GPUs, global batch size of 64, microbatch size of 4 per GPU, and gradient accumulation of 2. The impact of negative prompt numbers is discussed in Section 5.1, with additional details in Appendix F.
> - **Reliability of Perplexity** Perplexity may not fully reflect long-context performance. We placed the perplexity results in Appendix G, using the PG-19 dataset as a test set, not for validation during training.
> - **Inverse Trend of Perplexity and Performance** We respectfully disagree with the concern on perplexity and performance trends. Our analysis shows that minor perplexity differences have little impact on model results. The primary goal was to show that LOGO maintains its language modeling ability, and performance should be assessed using benchmarks like LongBench, not just perplexity.
>
> ---
>
> **[Question]** Lack of breakdown of evaluation details
>
> **[Re]** To provide clarity, we first present the partial context length distribution of LongBench. We can find that LongBench has a tightly concentrated length distribution, making it difficult to observe clear performance improvements across different length ranges.
>
> | Length Distribution         | 0-8K (%) | 8K-16K (%) | 16K+ (%) |
> |--------------------------------------|----------|------------|----------|
> | S-Doc QA                  | 66.0%   | 13.1%     | 20.9%   |
> | M-Doc QA    | 39.3%   | 60.3%     | 0.3%    |
> | Summ     | 62.5%   | 32.2%     | 5.3%    |
>
> We encourage the reviewer to refer to our additional experiments on LongBench V2 `(Reviewer icM3 Concern 2)` and Ruler. These benchmarks feature a broader and more diverse context length distribution. The results demonstrate that LOGO consistently improves the backbone model’s performance across all context lengths, with particularly significant gains at longer context lengths (128K), further validating LOGO's effectiveness for long context.
>
> ---
>
> We hope our responses have adequately addressed your concerns. If you have any further questions, please don’t hesitate to ask.

---

> > ### Comment · Reviewer_2obo · 2025-04-08
> >
> > The authors addressed my concerns and queries.
> >
> > I choose to retain my score.

---

> > > ### Author Response · Authors · 2025-04-09
> > >
> > > Thanks for your review again. We will surely add more experimental details and the breakdown of the evaluation results in the final revision.

---

### Official Review · Reviewer_icM3 · 2025-03-18

**Overall Recommendation:** 3

**Summary:**

The paper addresses the issue of long-context models struggling with generating coherent and accurate responses in real-world tasks.
It proposes LOGO, a preference optimization-based training strategy for long-context alignment, which includes efficient preference data synthesis and a reference-free training objective.
Experiments demonstrate improvements in multiple long-context tasks, with LOGO outperforming existing methods and maintaining or improving performance on short-context tasks.

**Claims And Evidence:**

The main claim of this paper that LCMs can achieve significant improvements in real-world tasks by training models with LOGO is not entirely rigorous, as an obvious issue is that increasing the relevant training data will always improve the model's effectiveness by training.

**Essential References Not Discussed:**

The references are comprehensive.

**Experimental Designs Or Analyses:**

I have checked the soundness of the experiments in this paper. For example the evaluation on LongBench introduced in Sec. 4.2. I think it is valuable to evaluate the proposed model on the newly released version LongBench-v2.

**Methods And Evaluation Criteria:**

It is sense-making.

**Other Comments Or Suggestions:**

The captions in Figure 1 should incorporate the benchmark (i.e., MMLU) used in the evaluation.

**Other Strengths And Weaknesses:**

Strengths
- I think the proposed training objective, which is similar to the SimPO, is useful to training LLMs with references optimization.
- The experiments are extensive to investigate the effectiveness of the model on various kinds of tasks.

Weakness
- As I mentioned above, the innovative contributions of this paper are limited. The proposed training object for LOGO seems like a variant of the SimPO. Further, the data construction pipeline is a widely used workflow.

**Questions For Authors:**

- In the section of Importance Scoring with Automatic Evaluator, how do you extract entities in both question and context? Why do these entities mater for measuring the importance? Why do these entities mater for measuring the importance of chunks?

**Relation To Broader Scientific Literature:**

The proposed training objective can contribute to the community for improving models on references corpus.

**Theoretical Claims:**

I have checked the proofs in this paper, which are reasonable. For example the proof that theoretical guarantee the synthetic positions can cover all possible scenarios listed in Appendix E.

---

> ### Author Rebuttal · Authors · 2025-03-29
>
> Dear Reviewer icM3, thanks for your insightful comments and suggestions. Below is our detailed response:
>
> ------
>
> **[Concern 1]** Rigor of the main claim that increasing training data generally improves model effectiveness
>
> **[Re]** We acknowledge that increasing high-quality training data can improve model performance, but training efficiency is a critical factor. Our work focuses on the efficient long-context alignment algorithm in LOGO rather than pure data scaling. Indiscriminate pure data scaling offers limited returns on long-context tasks. As shown in Table 1, SFT with 5B tokens (Data-Engineering-80K) improves LongBench performance by only +1.3 points, whereas LOGO achieves +2.5 points using just 0.3B tokens (16× less data). Additionally, Table 3 shows that LOGO improves performance (40.7 → 47.0), while SFT obtains a smaller gain (40.7 → 43.2) with a similar amount of data. Moreover, scaling data size in SFT requires careful balancing, as improper ratios (Figure 10) can hinder alignment.  `We also compare LOGO with other long-context DPO method, please refer to the second point of Reviewer 6VnN.`
>
> ------
>
> **[Concern 2]**  Evaluation results on LongBench-V2
>
> **[Re]** We have included the results in LongBench-v2 below.
>
> | Model                                | Overall        | Easy        | Hard        | Short        | Medium       | Long        |
> |--------------------------------------|--------------|------------|------------|------------|------------|------------|
> | Llama-3-8B-Instruct-80K-QLoRA-Merged | 10.3        | 9.4        | 10.9       | 11.1       | 11.2       | 7.4        |
> | + LOGO                               | 18.3 (**+8.0**) | 17.2 (**+7.8**) | 19.0 (**+8.1**) | 20.6 (**+9.5**) | 15.3 (**+4.1**) | 26.9 (**+19.5**) |
> | Mistral-7B-Instruct-v0.3             | 25.6        | 24.5       | 26.4       | 30.0       | 25.6       | 18.5       |
> | + LOGO                               | 29.8 (**+4.2**) | 30.2 (**+5.7**) | 29.6 (**+3.2**) | 35.0 (**+5.0**) | 28.4 (**+2.8**) | 26.9 (**+8.4**) |
> | Qwen2.5-7B-Instruct                  | 30.2        | 32.3       | 28.9       | 37.8       | 25.1       | 27.8       |
> | + LOGO                               | 32.8 (**+2.6**) | 35.9 (**+3.6**) | 30.9 (**+2.0**) | 40.6 (**+2.8**) | 28.8 (**+3.7**) | 33.3 (**+5.5**) |
>
> We observed that Llama-3-8B-Instruct-80K-QLoRA-Merged performed significantly worse than expected, with results even lower than the *25% random guessing baseline*, while LOGO can improve by an average of 8 points. We have carefully verified this outcome using the official code and double-checked our implementation to ensure its correctness.  After applying **LOGO training**, both models exhibit substantial performance gains, further validating the effectiveness of our approach.
> `Additionally, based on the suggestions from Reviewer 2obo, we have also provided the evaluation results on **Ruler** in the second point in the rebuttal box for Reviewer 2obo.`
>
> ------
>
> **[Concern 3]** Innovative Contributions
>
> **[Re]** Long-context capabilities have become indispensable, making effective long-context alignment more critical than ever. To our knowledge, **LOGO is among the first open-source RL methods tailored for long-context alignment during the time of this paper submission**, whereas prior works primarily achieve alignment with SFT. As demonstrated earlier, SFT alone yields limited benefits for long-context tasks, and while RLHF is a promising alternative, existing RL methods (e.g., DPO) are not well-suited for long-context alignment due to deployment challenges and inefficiencies.
>
> While SimPO is effective for short-context generation, it lacks adaptations for long-context tasks. LOGO introduces: 1) **Positional synthesis**, enabling efficient handling of long sequences. 2) **A novel preference data construction method**, along with a novel preference strategy, to address the lack of long-context evaluation models.
>
> ------
>
> **[Question 1]** Entity Extraction and Importance Scoring
>
> **[Re]**
>   1) **Entity Extraction**: As noted in Section 4.1 (Lines 261–262), we use spaCy to extract entities (e.g., person names, locations) from questions and chunks.
>   2) **Importance Metric**: In long-context QA tasks, questions often target specific snippets within the long context. Overlapping entities between a chunk and a question indicate higher relevance (e.g., a chunk mentioning "Berlin" is likely critical for a question about "Germany’s capital"). We empirically found that selecting top-K chunks (e.g., K=16 for 512-token chunks) suffices to cover salient information, balancing efficiency and accuracy. Therefore, we use the number of overlapping entities as a key metric to measure importance, and the importance score reflects the relevance of each chunk and the question.
>
> ------
>
> We hope our responses have adequately addressed your concerns. If you have any further questions, please don’t hesitate to ask.

---

### Decision · Program_Chairs · 2025-05-01

**Decision:**

Accept (poster)

**Comment:**

This paper introduces LOGO, a novel preference optimization method tailored for long-context alignment in LLMs. The method is efficient, reference-free, and shows consistent performance improvements across multiple long-context benchmarks with a modest training budget. While some aspects are incremental, the authors provide thorough empirical validation and address reviewer concerns well in the rebuttal. I lean toward recommending acceptance based on its practical contributions to long-context alignment.